# Sample-Efficient Alignment for LLMs

## Abstract

We study methods for efficiently aligning large language models (LLMs) with human preferences given budgeted online feedback. We first formulate the LLM alignment problem in the frame of contextual dueling bandits. This formulation, subsuming recent paradigms such as online RLHF and online DPO, inherently quests for sample-efficient algorithms that incorporate *online active exploration*. Leveraging insights from bandit theory, we introduce a unified algorithm based on **Thompson sampling** and highlight its applications in two distinct LLM alignment scenarios. The practical agent that efficiently implements this algorithm, named **SEA** (**S**ample-**E**fficient **A**lignment), is empirically validated through extensive experiments across three model scales (1B, 2.8B, 6.9B) and three preference learning algorithms (DPO, IPO, SLiC). The results demonstrate that **SEA** achieves highly sample-efficient alignment with oracle's preferences, outperforming recent active exploration methods for LLMs. We will release our codebase to hopefully accelerate future research in this field.

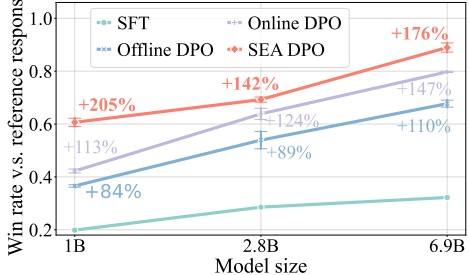 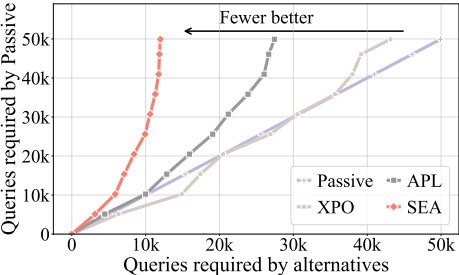

**Figure 1:** Win rate comparison of model responses against reference responses on the TL;DR task, judged by the preference oracle. All compared methods use the same optimization method (DPO). **(Left)** Performance improvements at convergence over SFT models achieved by offline (Offline DPO), passively online (Online DPO), and our *active exploration* (**SEA** DPO) methods. **(Right)** The number of queries required by the passively online method (Passive) versus that by different active exploration methods to attain various levels of win rates. **SEA** achieves the best sample efficiency for online alignment compared to XPO and APL.

## 1 Introduction

Aligning LLMs with human preferences is a crucial step to elicit various desirable behaviors, e.g., helpfulness and harmlessness [5]. Moreover, it holds the potential to create superhuman capabilities with only human-level feedback, as verifying is believed to be easier than synthesizing novel behaviors. By iteratively generating new candidates and asking for human feedback, LLMs could learn to reinforce good behaviors and may eventually surpass human capabilities.

Existing methods, either via reinforcement learning from human feedback (RLHF) [65, 50] or direct alignment from preferences (DAP) [55, 4], typically require a large amount of human annotations to achieve effective alignment. As a result, the volume of human feedback becomes a major bottleneck in practical alignment scenarios. This poses a challenging and under-explored research question:

Submitted to 39th Conference on Neural Information Processing Systems (NeurIPS 2025). Do not distribute.

To seek answers, in Sec. 2, we formalize LLM alignment as a contextual dueling bandit (CDB) [85, 20], where the agent (i.e., the learner and decision maker, in our case the LLM) interacts with the environment (i.e., human) to collect experience for improving its policy. This formulation naturally calls for two key properties for alignment algorithms to be sample-efficient:

**Property 1** (**Online interaction**). Interacting and learning *online* allows the agent to act with the latest learned policy and then use that experience to immediately improve the policy.

**Property 2** (**Active exploration**). An *actively exploring* agent strategically selects actions such that the collected experience leads to maximal policy improvement.

Since the CDB formulation is general and almost subsumes all existing LLM alignment methods, it provides us a lens to scrutinize prior methods on the axes of Properties 1 and 2. In Sec. 3, we thoroughly discuss prior alignment approaches, ranging from offline learning [55, 4] and passive learning with iterative [15, 18] or online interaction [24], to active exploration for learning preference models [21] or aligning LLMs [47, 86, 79]. As will be revealed, most prior methods (partially) fail to satisfy the two properties, resulting in inferior sample efficiency. Moreover, through the CDB formulation, we identify two LLM alignment scenarios, namely aligning from online users' feedback (e.g., ChatGPT [13]) and aligning from crowdsourcing [15, 50], and shed light on their correspondences to two bandit settings (explore & exploit and best arm identification). Understanding their differences is important for designing efficient alignment algorithms for respective scenarios. We detail these two settings in Sec. 2 and discuss how prior works approach them in Sec. 3.

Leveraging algorithmic insights from bandit theory, our answer to the research question above is a principled alignment algorithm based on Thompson sampling (TS) [71]. Our method fulfills Properties 1 and 2 to enhance sample efficiency, and it solves either of the two settings depending on practical scenarios (Sec. 4.1). We incorporate techniques including *epistemic reward model*, *policy-guided search* and *mixed preference learning* to implement the proposed TS algorithm (Sec. 4.2), yielding a practical agent which we call **SEA** (**S**ample-**E**fficient **A**lignment). In addition, we develop and will open source a highly efficient, distributed learning system for studying online LLM alignment methods (Sec. 5), eliminating barriers to *fair* empirical comparisons of different alignment algorithms. Through extensive experiments (Sec. 6), **SEA** shows strong empirical results (see Fig. 1), consistently achieving higher win rates and improved sample efficiency compared to baseline approaches across three model scales. We will open source the codebase to hopefully accelerate future research in this field.

In summary, the contributions of this work are:

- Through the lens of contextual dueling bandits, we propose a principled *Thompson sampling* algorithm for LLM online exploration, addressing both *explore & exploit* and *best arm identification* settings.

- We develop two novel techniques to approximate Thompson sampling in LLM's large action space: policy-guided search and mixed preference learning. Thompson sampling requires sampling a reward function from the posterior distribution and generating the sequence that maximizes the sampled reward function. For **policy-guided search**, we use an existing epistemic reward model for approximating the posterior and propose an approximate maximization method based on sampling a finite set of sequences from the LLM, and doing maximization on the finite sample. However, maintaining and updating a separate LLM for each reward function as suggested by Thompson sampling would be prohibitively expensive, thus **mixed preference learning** is introduced to align the LLM with internal reward functions to better approximate the maximization.

- To our knowledge, we are the first to study active exploration for LLM alignment with fully online experimental verification. The online alignment codebase will be open sourced to accelerate future studies.

## 2   LLM alignment as contextual dueling bandits

We first review the definitions and two typical objectives of *Contextual Dueling Bandits* (Sec. 2.1), then translate them into the language of *LLM alignment* (Sec. 2.2). The tight connection between them, as we will see, allows us to leverage insights from bandit algorithms to design efficient alignment algorithms for LLMs.

## 2.1 Contextual dueling bandits

Contextual dueling bandits (CDB) [85, 20] is proposed to study online learning problems where the feedback consists of relative pairwise comparisons. A CDB problem can be characterized by a tuple $(\mathcal{C}, \mathcal{A}, \mathbb{P})$, where $\mathcal{C}$ is the context space, $\mathcal{A}$ is the action space, and $\mathbb{P} : \mathcal{A} \times \mathcal{A} \times \mathcal{C} \mapsto [0, 1]$ denotes the unknown *preference oracle*. An agent learns by iteratively interacting with the environment (i.e., the preference oracle $\mathbb{P}$) as follows. At each round $t$ of the learning process, a context $c_t \sim p_{\mathcal{C}}$ is presented to the agent, who needs to take two actions $a_t, a'_t \in \mathcal{A}$ for a "dueling" comparison. The agent then receives stochastic feedback in the form of a comparison result $z_t \sim \text{Ber}\left(\mathbb{P}\left(a_t \succ a'_t | c_t\right)\right)$ from the environment, where $\text{Ber}(\cdot)$ is the Bernoulli distribution and $\succ$ denotes that the first action is preferred.

**Regret**. The quality of the dueling actions selected by the agent is measured by the *immediate regret*: $R_t = \mathbb{P}(a_t^\star \succ a_t | c_t) + \mathbb{P}(a_t^\star \succ a'_t | c_t) - 1$, where $a_t^\star$ is the best action[1] the agent would take at round $t$ if it had complete knowledge of $\mathbb{P}$. Intuitively, if the agent has learned how to act optimally from round $t$ onwards, it would no longer suffer any regret since its actions would be indistinguishable from the best action ($\mathbb{P}(a_\tau^\star \succ a_\tau | c_\tau) = \frac{1}{2}$ hence $R_\tau = 0$ for $\tau \geq t$).

**Optimal policy**. A policy $\pi \in \Delta_{\mathcal{A}}^{\mathcal{C}}$[2] associates each context $c \in \mathcal{C}$ with a probability distribution $\pi(\cdot | c) \in \Delta_{\mathcal{A}}$ over the action space. The *total preference* of policy $\pi$ over policy $\mu$ given a context sampling distribution $p_{\mathcal{C}} \in \Delta_{\mathcal{C}}$ and a preference oracle $\mathbb{P}$ is defined as

$$P_{p_{\mathcal{C}}, \mathbb{P}}(\pi \succ \mu) = \mathbb{E}_{c \sim p_{\mathcal{C}}}\left[\mathbb{E}_{a \sim \pi(\cdot | c)} \mathbb{E}_{a' \sim \mu(\cdot | c)}\left[\mathbb{P}(a \succ a' | c)\right]\right]. \tag{1}$$

We adopt the *von Neumann winner* [20] as the solution concept, which requires the optimal policy $\pi^\star$ to satisfy that

$$\forall \pi' \in \Delta_{\mathcal{A}}^{\mathcal{C}}, \ P_{p_{\mathcal{C}}, \mathbb{P}}(\pi^\star \succ \pi') \geq \frac{1}{2}. \tag{2}$$

In words, the von Neumann winner policy should beat or tie with every policy (i.e., is zero-regret) on average.

**Learning objectives.** The goal of bandit agents is to learn an optimal policy through interactions with the environment. There are two subtypes of objectives that focus on different learning scenarios. The first type considers the conventional *explore and exploit (E&E)* setting [59, 3], where the agent learns fully **online** and tries to minimize the cumulative regret over $T$ rounds: $\sum_{t=1}^{T} R_t$. The second type of objective concerns the *best arm identification (BAI)* setting [9, 2], where the agent is only evaluated **offline** on its average performance, possibly at any round (a.k.a., anytime regret), and tries to learn the optimal policy with minimum interaction. Both settings call for effective *online exploration* strategies that satisfy Properties 1 and 2. Their differences will be made clearer with real scenarios in Sec. 2.2.

## 2.2 Online alignment as CDB

Online LLM alignment can be framed as a CDB problem. Specifically, at time $t$ a text prompt (*cf.* context) $x_t \in \mathcal{X}$ is sampled from a prompt distribution $p_{\mathcal{X}}$. Then, two distinct responses (*cf.* actions), $y_t, y'_t \in \mathcal{Y}$, are chosen by the agent, and presented to human annotators (*cf.* the environment) for preference ranking. The winning and losing responses are labeled as $(y_t^+, y_t^-)$ based on a binary stochastic feedback $z_t \sim \text{Ber}\left(\mathbb{P}\left(y_t \succ y'_t | x_t\right)\right)$. The agent is expected to produce *good* responses satisfying either E&E or BAI objectives, with knowledge learned from the experience accumulated so far: $\mathcal{D}_t = \{(x_\tau, y_\tau^+, y_\tau^-)\}_{\tau=1}^{t}$. A standard assumption is that human preferences follow the Bradley-Terry (BT) model [8]:

$$\mathbb{P}(y_t \succ y'_t | x_t) = \frac{\exp\left(r^\star(x_t, y_t)\right)}{\exp\left(r^\star(x_t, y_t)\right) + \exp\left(r^\star(x_t, y'_t)\right)} = \sigma(r^\star(x_t, y_t) - r^\star(x_t, y'_t)), \tag{3}$$

where $\sigma$ is the sigmoid function and $r^\star$ encodes human's implicit reward. The immediate regret of LLM alignment can be rewritten as $R_t = r^\star(x_t, y_t^\star) - \left(r^\star(x_t, y_t) + r^\star(x_t, y'_t)\right)/2$ with the BT assumption [62, 39], where $y_t^\star$ is the best response for prompt $x_t$ given human's implicit reward, i.e., $r^\star(x_t, y_t^\star) \geq r^\star(x_t, y), \forall y \in \mathcal{Y}$. The von Neumann winner policy is also redefined as

$$\pi^\star \in \underset{\pi \in \Delta_{\mathcal{Y}}^{\mathcal{X}}}{\arg \max} J(\pi), \ \text{where } J(\pi) = \mathbb{E}_{x \sim p_{\mathcal{X}}} \mathbb{E}_{y \sim \pi(\cdot | x)}[r^\star(x, y)] \text{ is the objective}, \tag{4}$$

---

[1]We assume that a best action $a^\star$ in the sense that $\mathbb{P}(a^\star \succ a | c) \geq \frac{1}{2}, \forall a \in \mathcal{A}$ exists for all context $c \in \mathcal{C}$.

[2]We denote by $\Delta_{\mathcal{A}}^{\mathcal{C}}$ the set of all mappings $\mathcal{C} \mapsto \Delta_{\mathcal{A}}$, where $\Delta_{\mathcal{A}}$ denotes the set of all probability distributions over $\mathcal{A}$.

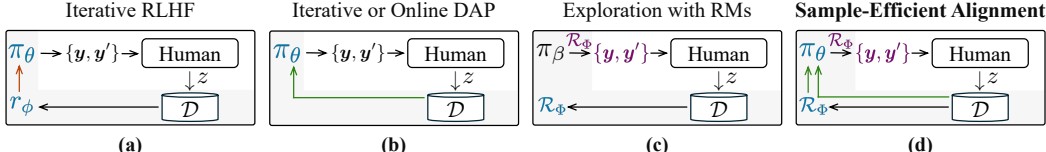

**Figure 2:** Different paradigms to solve online LLM alignment in the CDB interface. The CDB agent is shaded in gray. We use colors to denote learnable components, RL optimizer, direct optimizer, and active exploration. $r_\phi$ denotes a point estimate of human's implicit reward, while $\mathcal{R}_\Phi$ refers to an uncertainty-aware reward model. Please see Sec. 3 for detailed comparisons with references to prior works. [fdaL: updated fig.3 to highlight the differences between (b) and (d).]

by substituting Eq. (3) into Eq. (1) and maximizing $P_{p_\mathcal{X}, \mathbb{P}}(\pi \succ \pi^\star)$ towards $1/2$.

The **two settings in bandits** have their respective applications in LLM alignment. **(1)** The E&E setting applies to the scenario of serving an LLM-based application online and aligning it continually with users' preferences. In this setting, the agent needs to balance exploration with exploitation, thus the cumulative regret is of interest because the quality of *every* response matters. In fact, commercial systems like ChatGPT would strategically ask users to make a dueling comparison, while upholding the quality of both responses. Please see Fig. 11 in App. I for an example. **(2)** The BAI setting corresponds to the other scenario where annotators are paid to provide human feedback [15, 50]. The desideratum in this scenario is to align the LLM at the minimum labeling cost, while the quality of the dueling responses is not important as long as the experience helps sample-efficiently learn the von Neumann winner policy.

After formalizing LLM alignment in the framework of CDB and uncovering their tight connections, we next thoroughly discuss existing alignment methods in the CDB framework and reveal the sources of their sample inefficiencies.

# 3 How prior works (partially) solve LLM alignment as CDB

We first align the notations and terminology used in CDB with commonly referred ones in the LLM community. Previously, we used the term "agent" to denote the learner and decision maker, and referred to its overall behavior as the "policy" $\pi$ (as in Eq. (4)), following the standard abstraction in RL [67, 68]. However, in the LLM literature, "policy" typically refers to the generative language model alone, excluding components like reward models (RMs) that the agent might additionally build. To avoid confusion, from now on we use $\pi_{\theta^t}$ to denote the generative language model (policy) and $r_{\phi^t}$ to denote the (optional) RM at time $t$, both of which are learned from preference data $\mathcal{D}_t$ collected up to time $t$. We will omit $t$ when the time-indexing is not applicable (i.e., no online interaction) or not important in the context.

**RLHF and DAP**. Commonly adopted RLHF pipelines [15, 65, 5, 50] first learn a proxy RM with a negative log-likelihood loss:

$$\mathcal{L}_r(\phi|\mathcal{D}) = -\mathbb{E}_{(\boldsymbol{x}, \boldsymbol{y}^+, \boldsymbol{y}^-) \sim p_\mathcal{D}} \left[ \log \sigma \left( r_\phi\left(\boldsymbol{x}, \boldsymbol{y}^+\right) - r_\phi\left(\boldsymbol{x}, \boldsymbol{y}^-\right) \right) \right], \tag{5}$$

where $\mathcal{D}$ is collected by querying human annotators using a behavior policy $\pi_{\mathrm{ref}}$ (typically the supervised fine-tuned policy $\pi_{\mathrm{sft}}$). Afterwards, *offline RL*[3] [36, 37] is conducted to learn $\pi_\theta$ with respect to the learned reward $r_\phi$ internally within the agent (Fig. 2a). However, the learned model $\pi_\theta$ might be inaccurate at regions out of the distribution (o.o.d.) of $\pi_{\mathrm{ref}}$ because little training data can be collected. An effective remedy is to incorporate a pessimistic term to combat the distributional shift, leading to a reformulation of the von Neumann winner policy objective in Eq. (4) as

$$J(\pi_\theta) = \mathop{\mathbb{E}}_{\boldsymbol{x} \sim p_\mathcal{X}} \mathop{\mathbb{E}}_{\boldsymbol{y} \sim \pi_\theta(\cdot|\boldsymbol{x})} \left[ \underbrace{r_\phi(\boldsymbol{x}, \boldsymbol{y})}_{\text{estimated } r^\star} - \underbrace{\beta \log \frac{\pi_\theta(\boldsymbol{y}|\boldsymbol{x})}{\pi_{\mathrm{ref}}(\boldsymbol{y}|\boldsymbol{x})}}_{\text{o.o.d. reward penalty}} \right] \tag{6}$$

$$= \mathop{\mathbb{E}}_{\boldsymbol{x} \sim p_\mathcal{X}} \left[ \mathop{\mathbb{E}}_{\boldsymbol{y} \sim \pi_\theta(\cdot|\boldsymbol{x})} [r_\phi(\boldsymbol{x}, \boldsymbol{y})] - \beta D_{\mathrm{KL}}(\pi_\theta(\cdot|\boldsymbol{x})||\pi_{\mathrm{ref}}(\cdot|\boldsymbol{x})) \right], \tag{7}$$

which converts an online objective regarding the human's implicit reward $r^\star$ to an offline objective regarding the proxy reward $r_\phi$. The KL penalty in Eq. (15) is widely used for language model

---

[3]*Offline* in the sense that $\pi_\theta$ is not directly learned from online human feedback. See App. C for details.

fine-tuning [29, 80], and PPO [64] has become a default *RL optimizer* to maximize the KL-regularized reward. However, the performance of RLHF is guaranteed only if the preference data $\mathcal{D}$ induced by $\pi_{\text{ref}}$ adequately covers $\pi^\star$ [90], which is often approximated by updating $\pi_{\text{ref}}$ with the latest (improved) $\pi_\theta$ for re-sampling a batch of online experience and repeating Eq. (13) and (15). Prior works typically focus on offline or iterative online (with only a few iterations) settings [80, 18], which may compromise sample efficiency (Property 1).

True online RLHF is difficult due to the complexity and instability of RL optimizers. For example, Huang et al. [27] openly reproduces offline RLHF scaling behaviors but requires many implementation tricks for training, highlighting the difficulties of an online counterpart. Fortunately, the introduction of DAP (or *direct optimizers*) largely simplifies and stabilizes fine-tuning by conducting contrastive supervised learning directly on $\mathcal{D}$ (Fig. 2b). While most DAP works focus on learning from a fixed offline preference dataset, including Zhao et al. [88], Rafailov et al. [55], Azar et al. [4], Meng et al. [46], Zhang et al. [87]), iterative DPO [81] observes improved results when allowing iterative online interaction. Guo et al. [24] further propose OAIF to make DAP faithfully online, satisfying Property 1, and demonstrate that online learning prevents over-fitting and yields continual performance improvement. Nevertheless, it still employs passive exploration strategies (using $\boldsymbol{y}, \boldsymbol{y}' \sim \pi_\theta$), hindering sample efficiency (Property 2).

**Online exploration in LLMs**. A line of recent works [44, 17, 45, 21] adopts the fully online bandit formulation and incorporates *active exploration* with uncertainty-aware RMs for response selection (Fig. 2c). In particular, Mehta et al. [44] consider the E&E setting and develop a UCB-style [3] algorithm; Das et al. [17] instead select the dueling responses with the most uncertain preference estimate, targeting the BAI setting in a pure exploration way; unlike the above, Melo et al. [45] view the problem from the angle of pool-based active learning and propose an acquisition function based on both entropy and epistemic uncertainty; finally, the work by Dwaracherla et al. [21] is the closest to ours in the sense that they apply double Thompson sampling (DTS) [78] for exploration, but DTS is designed for the E&E setting while they evaluate anytime average performance as in the BAI setting. We will show in App. G.1 that pure exploration by Das et al. [17] is not the best choice for BAI, and the objective mismatch in Dwaracherla et al. [21] could lead to suboptimal performance in respective settings. Meanwhile, all these works primarily focus on learning uncertainty-aware RMs online without updating LLM policies. Therefore, all responses are sampled from a fixed proposal policy $\pi_\beta$ (or even a fixed dataset), making the data coverage a critical concern.

Another line of research updates LLMs online while incorporating exploration. Zhang et al. [86] and Xie et al. [79] independently propose to learn an optimistic RM to encourage exploration. They leverage the property of DPO [55] to reparameterize RM with policy and conclude with an extra optimistic term in the DPO loss function. Thus, their learning processes are like Fig. 2b but with an optimistic direct optimizer. Muldrew et al. [47] adopt the vanilla DPO loss but utilize the implicit reward margin to actively select dueling responses. Yet, these methods are tightly coupled with DPO and not compatible to other direct optimizers. Their experiments are also limited to a few online iterations, possibly due to the implementation difficulty of a faithfully online learning system. Given their relevance to our approach, we will reproduce them in a fully online manner for fair comparisons in Sec. 6.1. We summarize prior works in Table 2 in App. I.

## 4  **SEA: sample-efficient alignment for LLMs**

In this section we present our online exploration agent **SEA** (Fig. 2d). We first introduce a principled Thompson sampling algorithm inspired by bandit theory (Sec. 4.1), and then derive **SEA** as its practically efficient implementation (Sec. 4.2). Interestingly, **SEA** can also be viewed as an instantiation of a classical model-based RL architecture called Dyna [66], for which we defer the discussion to App. C.

### 4.1  **Thompson sampling for LLM alignment**

**Thompson sampling (TS)** [71] is widely adopted for solving bandit problems at scale due to its efficiency and strong empirical performance in general online learning problems [12, 61]. A bandit agent using Thompson sampling typically maintains and incrementally updates a posterior distribution of the oracle reward $p(r|\mathcal{D})$. Meanwhile, the agent takes actions following a greedy policy with respect to a sampled RM: $\boldsymbol{a}_t = \arg\max_{\boldsymbol{a}} r(\boldsymbol{a})$ with $r \sim p_r(\cdot|\mathcal{D})$. This simple yet effective algorithm naturally balances exploration and exploitation: when the agent has limited knowledge about the environment, the posterior estimate exhibits high uncertainty so that the sampled RM could guide the

---

**Algorithm 1** Thompson sampling for LLM alignment (intractable).

**Input:** Prompt distribution $p_{\mathcal{X}}$, unknown but queryable preference oracle $\mathbb{P}$.

1: Initialize experience $\mathcal{D}_0 \leftarrow \varnothing$.
2: **for** $t = 1, \ldots, T$ **do**
3:     Receive a prompt $\boldsymbol{x}_t \sim p_{\mathcal{X}}$.
4:     Sample $r \sim p_r(\cdot|\mathcal{D}_{t-1})$ and set $\boldsymbol{y}_t \leftarrow \arg\max_{\boldsymbol{b} \in \mathcal{Y}} r(\boldsymbol{x}_t, \boldsymbol{b})$.          `// Select 1st response y.`
      `// E&E objective: aligning an online system.`
5:     **repeat**
        Sample $r \sim p_r(\cdot|\mathcal{D}_{t-1})$ and set $\boldsymbol{y}'_t \leftarrow \arg\max_{\boldsymbol{b} \in \mathcal{Y}} r(\boldsymbol{x}_t, \boldsymbol{b})$.     `// Select 2nd response y'.`
        **until** $\boldsymbol{y}'_t \neq \boldsymbol{y}_t$
      `// BAI objective: labeling via crowdsourcing.`
6:     Set $\boldsymbol{y}'_t \leftarrow \arg\max_{\boldsymbol{b} \in \mathcal{Y}} \mathbb{V}\left[\sigma\left(r(\boldsymbol{x}_t, \boldsymbol{y}_t) - r(\boldsymbol{x}_t, \boldsymbol{b})\right)\right]$,      `// OR select 2nd response y'.`
        where $\mathbb{V}[\cdot]$ computes variance over the posterior $p_r(\cdot|\mathcal{D}_{t-1})$.
7:     Query $\mathbb{P}$ to label $\{\boldsymbol{y}_t, \boldsymbol{y}'_t\}$, and update experience $\mathcal{D}_t \leftarrow \mathcal{D}_{t-1} \bigcup \{(\boldsymbol{x}_t, \boldsymbol{y}^+_t, \boldsymbol{y}^-_t)\}$.
8: **end for**

      `// See Algorithm 2 for a practical version.`

---

greedy policy to explore; after sufficient experience is gathered, the sampled RM approximates the oracle more closely, allowing the agent to exploit near-optimal policies.

In the context of LLM alignment, we leverage the BT assumption (Eq. (3)) to replace the preference oracle $\mathbb{P}$ with human's implicit reward $r^\star$. This substitution enables us to model the reward posterior $p(r|\mathcal{D})$ in the standard TS framework, preserving the probabilistic structure necessary for effective posterior sampling. Inspired by prior works [78, 23] on non-contextual $K$-arm bandits and preferential Bayesian optimization problems, we generalize them for LLM alignment and develop a unified algorithm as shown in Algorithm 1. Note that we assume for now the LLM agent can be fully described by the posterior $p(r|\mathcal{D})$, and we defer practical reward ($r_\phi$) and policy ($\pi_\theta$) learning to Sec. 4.2.

As Algorithm 1 presents, the first response of the duel is always selected via standard TS (Line 4). The selection of the second response varies across different settings. Line 5 will be used for scenarios where preference feedback is collected from online users (the E&E setting). The dueling responses selected in this case will both try to maximize a sampled RM, so that the online user experience is warranted with best effort. However, such algorithm can have poor asymptotic performance for BAI problems [60], because sub-optimal responses with confidently high rewards might be tried for a long time at the expense of not exploring other potentially better choices. In light of this, Line 6 provides an alternative for scenarios where we could hire annotators for feedback and low-quality but exploratory responses are safe to try. Specifically, Line 6 selects the second response as the one that maximizes the variance of the preference (Eq. (3)) over the first response $\boldsymbol{y}_t$. This variance quantifies the *epistemic uncertainty* of the RM, pointing the agent to the maximally informative direction to explore for better sample efficiency.

However, Algorithm 1 is yet to be practical for LLM alignment for three main reasons. First, computing and sampling from a reward posterior is intractable for nearly all RMs at LLM scale, which are mostly based on large transformers [35]. Second, even if we managed to approximate the reward posterior, the $\arg\max$ operations for response selection are still intractable since the search space $\mathcal{Y}$ is discrete and massive for token sequences of arbitrary length. Last but not least, an LLM agent [1, 72] typically consists in a generative model $\pi_\theta$ (e.g., a transformer [73]), while the algorithm above is centered around a reward posterior $p(r|\mathcal{D})$ that cannot be easily converted into a generative model. We next detail how **SEA** practically addresses the three aforementioned issues.

## 4.2 Practical implementation

### 4.2.1 Epistemic reward model for posterior sampling

To implement active exploration with TS, we seek an efficient way to maintain and incrementally update the reward posterior $p(r|\mathcal{D})$. We consider *deep ensemble* for our purpose, due to its capability to model epistemic uncertainty [34] and provable results when applied to TS in linear bandits [54]. Specifically, we update a set of plausible RMs independently and online, using the preference data

and a regularized negative log-likelihood loss:

$$\mathcal{L}_{\mathcal{R}}(\Phi^t | \mathcal{D}_t) = \sum_{k=1}^{K} \left( \mathcal{L}_r(\phi_k^t | \mathcal{D}_t) - \lambda ||\phi_k^t - \phi_k^0|| \right), \tag{8}$$

where $\mathcal{L}_r$ is defined in Eq. (13), $\Phi^t = \{\phi_k^t\}_{k=1}^K$ contains the weights of the ensemble of size $K$, and $\lambda$ controls the regularization towards individual initial weights $\phi_k^0$. Each ensemble member is initialized independently with random weights, and then trained with regularization to maintain the diversity across ensemble members [21]. Randomly picking a $\phi_k^t$ from $\Phi^t$ would approximate the posterior sampling ($r \sim p_r(\cdot|\mathcal{D}_t)$) for the RM [43, 25]. In practice, we train $K$ MLP heads on top of a pretrained and frozen transformer. We refer to the ensemble as the Epistemic Reward Model (ERM, denoted as $\mathcal{R}_\Phi$).

### 4.2.2 Policy-guided search to approximate $\arg\max$

With the ERM approximating the reward posterior, we need to further approximate the response selection steps (Lines 4 to 6) which generally take the form of $\arg\max_{\boldsymbol{b} \in \mathcal{Y}} U(\boldsymbol{b})$, where $U$ absorbs the sampled prompt, the sampled RM, and optionally the selected first response (for BAI, Line 6). To obtain the maximum, bandit algorithms for large action spaces typically resort to an action optimization oracle [31, 91], but they assume a linear structure of $U$ with respect to $\boldsymbol{b}$, which might be impractical for LLMs. Therefore, we instead replace the optimization over $\mathcal{Y}$ with sampling from a policy-guided distribution conditioned on $U$, $\pi_{\mathrm{prior}}(\cdot|\boldsymbol{x}) \exp(U(\cdot)/\eta)$, which is appropriate since it favors responses $\boldsymbol{y}$ that approximately maximize $U(\boldsymbol{y})$. In practice, for a given prompt $\boldsymbol{x}_t$, we sample $M$ candidate responses from the prior policy $\pi_{\mathrm{prior}}(\cdot|\boldsymbol{x}_t)$ to construct a proposal set $\mathcal{S}_t = \{\boldsymbol{y}_t^i\}_{i=1}^M$. We then conduct a greedy search in $\mathcal{S}_t$ (taking $\eta \to 0$) to identify the response $\boldsymbol{y}_t$ (or $\boldsymbol{y}_t'$) that locally maximizes the utility function $U$, which is subsequently used in the duel. We also reuse the same $\mathcal{S}_t$ for different $U$ functions at time $t$ to save computation. The choice of $\pi_{\mathrm{prior}}$ will be discussed in the next section.

### 4.2.3 Online policy learning from mixed preferences

We finally resolve two remaining questions: *(Q1)* how to choose a sensible $\pi_{\mathrm{prior}}$ at each time $t$ and *(Q2)* how to get a good generative policy online. To this end, we propose a simple approach to approximately address both questions simultaneously. That is, we can utilize any direct optimizer to learn the policy $\pi_{\theta^t}$ online with the following loss and use the latest online policy as $\pi_{\mathrm{prior}}$:

$$\mathcal{L}_{\pi}(\theta^t | \mathcal{B}_t, \pi_{\mathrm{ref}}, F) = \mathbb{E}_{(\boldsymbol{x}, \boldsymbol{y}^+, \boldsymbol{y}^-) \sim p_{\mathcal{B}^t}} \left[ F_{\theta^t}(\boldsymbol{x}, \boldsymbol{y}^+, \boldsymbol{y}^-, \pi_{\mathrm{ref}}) \right], \tag{9}$$

where $\mathcal{B}_t$ is a batch of preference data labeled by the oracle wherein the responses are proposed by $\pi_{\mathrm{prior}}$ and selected by $\mathcal{R}_{\Phi^t}$, $F$ could be any DAP loss (see App. A for some examples), and $\pi_{\mathrm{ref}}$ is chosen to be $\pi_{\mathrm{sft}}$. Note that we use $\pi_{\theta^t}$ as $\pi_{\mathrm{prior}}$ at any time $t$, thus $\mathcal{B}^t$ is a batch of on-policy data. By *contrastive training* on these *on-policy* data, we leverage their orthogonal benefits to achieve maximal policy improvement [69, 70].

Now that optimizing Eq. (9) yields a good online policy $\pi_{\theta^t}$ (answering Q2), we need to assess whether $\pi_{\theta^t}$ can serve as a suitable $\pi_{\mathrm{prior}}$ for approximating the $\arg\max$ in TS (Q1). If we optimize $\pi_{\theta^t}$ with oracle preference data, $\mathcal{S}_t$ will be biased towards responses with high oracle reward $r^\star$. Bias towards high-$r^\star$ region is generally helpful because it aligns with $\arg\max_{\boldsymbol{b} \in \mathcal{Y}} r(\boldsymbol{x}, \boldsymbol{b})$ that seeks high-reward responses. However, optimizing $\pi_{\theta^t}$ *only* with oracle data can average out the epistemic uncertainty of $\mathcal{R}$, hindering the exploration efficiency. To mitigate this issue, we further align $\pi_{\theta^t}$ with $\mathcal{R}_{\Phi^t}$ using the same direct optimizer to encourage $\pi_{\theta^t}$ to propose high-$r_{\phi_k^t}$ responses for individual $r_{\phi_k^t}$, leading to better approximation of $\arg\max_{\boldsymbol{b} \in \mathcal{Y}} r(\boldsymbol{x}, \boldsymbol{b})$ for any sampled $r$. To implement, we optimize Eq. (9) over a batch of data mixture $p_{\mathcal{B}_t^{\mathrm{mix}}} = \gamma p_{\mathcal{B}_t} + (1-\gamma) p_{\mathcal{B}_t^{\mathrm{ERM}}}$, where $\gamma \in [0, 1]$ controls the mixture ratio and $\mathcal{B}_t^{\mathrm{ERM}} = \{(\boldsymbol{x}_i, \tilde{\boldsymbol{y}}_i^+, \tilde{\boldsymbol{y}}_i^-)\}_{i=1}^b$ consists of preference data labeled by randomly sampled individual ensemble members $r_{\phi_k^t}$. Interestingly, learning from mixed preferences further boosts sample efficiency because it utilizes the internal ERM to get pseudo labels instead of querying humans. This relates closely to model-based RL, for which we discuss further in App. C. We summarize our practical algorithm (Algorithm 2) in App. A.

## 5 Experimental setup

**Software**. To facilitate our empirical studies, we develop a distributed learning framework for online LLM alignment. The framework is based on an Actor-Learner-Oracle architecture, drawing

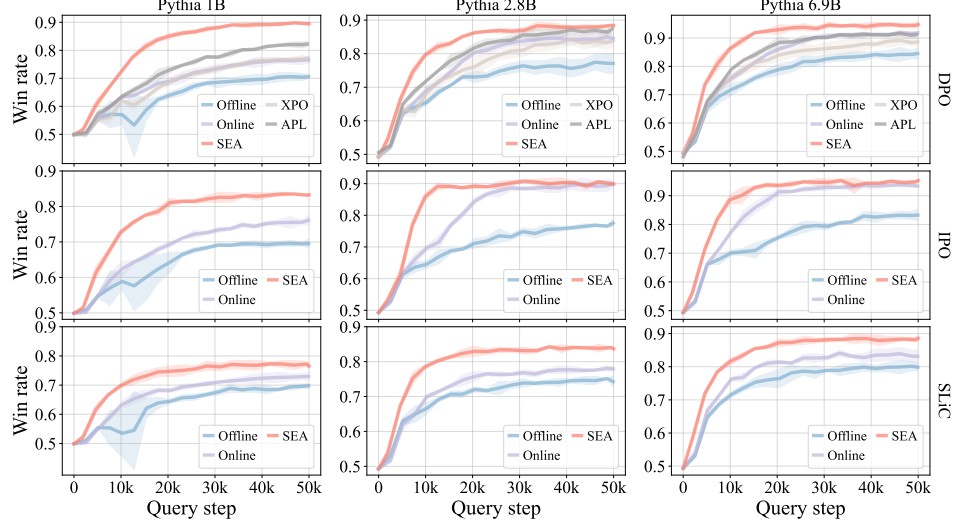

**Figure 3:** Win rate comparison of different algorithms against their initial SFT models across three scales and three direct optimizers.

inspiration from Espeholt et al. [22]. We incorporate various optimizations for each component: vLLM [33] for actors, DeepSpeed [58] for learners, and Mosec [83] for oracles. Detailed descriptions of the framework and its efficiency benchmarks are provided in App. D & H.

**Settings.** We adopt SFT models tuned on `TL;DR` [65] from Huang et al. [27], which cover three scales (1B, 2.8B, 6.9B) of the Pythia family [7], as starting points for our experiments. We use a strong scalar RM [40][4] to simulate the preference oracle. To verify the effectiveness of **SEA**, we employ three direct optimizers: DPO [55], IPO [4], and SLiC [88] to serve as $F$ in Eq. (9). Besides, two LLM exploration methods built on DPO, APL [47] and XPO [79], are fairly compared when using DPO as the optimizer. Our experiments primarily focus on the BAI setting (crowdsourcing labeling), where we report the win rate of learned models against initial SFT models. All experiments are repeated three times to ensure statistical significance. Please see App. F for more details.

## 6 Empirical studies

We next present our empirical studies highlighting five results: **(1)** Comparisons with baselines across various direct optimizers and model scales demonstrate **SEA**'s superior *sample efficiency* (Sec. 6.1). **(2)** Ablations confirm that both *online policy learning* and *active exploration* contribute to sample-efficient alignment, and using the learned ERM for Best-of-N sampling further improves the performance (Sec. 6.2). **(3)** Different exploration strategies (Line 5 or Line 6 in Algorithm 1) are verified to work best in respective settings. **(4)** **SEA** robustly outperforms baselines when GPT4o-mini is used as a judge to simulate human feedback. **(5)** Beyond the summarization task, **SEA** can effectively enhance general capabilities of LLMs. Results for (3-5) are deferred to App. G due to space constraints.

### 6.1 Overall comparison

We first compare **SEA** with all baselines across three model scales and three direct optimizers. APL and XPO are only compared when DPO is used as the direct optimizer, because they are incompatible with IPO or SLiC. Fig. 3 shows the win rate curves versus the number of query steps. Across all settings, Online agents consistently improve sample efficiency over their Offline counterparts, validating the necessity of **Property 1** for alignment algorithms. Focusing on the first row, we observe that among prior active exploration methods, XPO gives a small improvement in final performance over Online (passive) at the 1B scale, but falls short for larger scales. On the other hand, APL shows a significant sample efficiency boost at the 1B scale, but this advantage diminishes when scaling up and it performs almost the same as Online at 6.9B scale. Our method, **SEA**, outperforms both offline and online passive methods across all scales and all direct optimizers, confirming the critical role that **Property 2** plays for sample-efficient alignment. Meanwhile, in the special case of using DPO as the direct

---

[4]https://huggingface.co/Skywork/Skywork-Reward-Llama-3.1-8B.

**Table 1:** Decomposition of different driving factors of online active alignment algorithms.

| Variant | Inference (Test) | Exploration | Learn | Remark |
|---|---|---|---|---|
| 1 | $\pi_\theta$ | passive | $\pi_\theta$ | Online DAP [24] |
| 2 | $\pi_\theta$ | active | $(\pi_\theta, \mathcal{R}_\Phi)$ | **SEA** *without* ERM sync (Sec. 4.2.3) |
| 3 | $\pi_\theta$ | active | $(\pi_\theta \leftrightarrow \mathcal{R}_\Phi)$ | **SEA** |
| 4 | $\mathrm{BoN}(\pi_\theta, \mathcal{R}_\Phi)$ | passive | $(\pi_\theta, \mathcal{R}_\Phi)$ | - |
| 5 | $\mathrm{BoN}(\pi_\theta, \mathcal{R}_\Phi)$ | active | $(\pi_\theta, \mathcal{R}_\Phi)$ | - |
| 6 | $\mathrm{BoN}(\pi_\theta, \mathcal{R}_\Phi)$ | active | $(\pi_\theta \leftrightarrow \mathcal{R}_\Phi)$ | **SEA** with Best-of-N sampling |
| 7 | $\mathrm{BoN}(\pi_{\mathrm{ref}}, \mathcal{R}_\Phi)$ | active | $\mathcal{R}_\Phi$ | Not learn policy [21] |

optimizer, **SEA** also shows superior performance to prior online active exploration methods including APL and XPO. We invite readers to revisit Fig. 1, where we show that **SEA** not only attains significantly improved final performance (Top) but also achieves $2$-$5\times$ better sample efficiency (Bottom).

Additionally, we note that the choice of direct optimizer is crucial for both online learning and active exploration. When comparing different optimizers at the 1B scale (the first column), all Offline agents demonstrate comparable learning efficiency and reach the same level of final performance (around $70\%$ win rate), but SLiC Online agent deliver slightly less improvement than DPO and IPO Online agents. Besides, when incorporating active exploration, the **SEA** agent using DPO shows much larger improvement than the other two. This suggests that selecting the most suitable policy optimizer coupled with active exploration would yield the best agent.

### 6.2 Ablation analysis

We decompose **SEA** into distinct components to evaluate their individual contributions. Table 1 shows the three axes we dissect **SEA** on, including inference methods, exploration strategies, and learning components. We construct seven agent variants from different combinations, which cover two closely related baselines [21, 24]. We show in Fig. 4 the performance curves of each variant, all trained with DPO on 1B scale.

The top plot compares variants that directly use the policy for inference. Comparing with the vanilla online method (Variant-1), we observe learning ERM for active exploration (Variant-2) is beneficial, and aligning $\pi_{\theta^t}$ with $\mathcal{R}_{\Phi^t}$ (Variant-3) further improves sample efficiency, which validate our algorithm. Additionally, since a reward model is learned within the agent, we can incorporate inference-time alignment via Best-of-N (BoN) sampling [48, 72]. This also facilitates a direct comparison between **SEA** and Dwaracherla et al. [21], which learns a similar ERM for both exploration and BoN but does not align the LLM policy. Results in the bottom plot of Fig. 4 suggest a similar trend that Variant-6 $\succ$ Variant-5 $\succ$ Variant-4. The Variant-7 [21], however, ceases to improve after ERM converges due to the limited capability of its fixed policy.

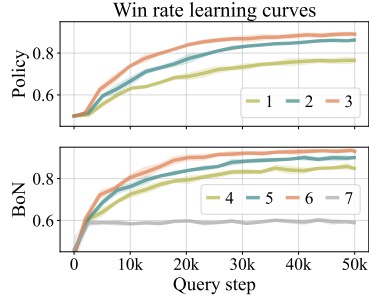

**Figure 4:** Win rate comparison of different agent variants when using (**Left**) policy and (**Right**) Best-of-N sampling for inference.

## 7 Conclusion

In this paper, we study the problem of LLM alignment through the lens of contextual dueling bandits and propose a Thompson sampling-based algorithm to achieve sample-efficient alignment. We incorporate three techniques, including epistemic reward model, policy-guided search and mixed preference learning to yield a practically efficient online alignment method. Extensive empirical evaluation demonstrates the superior sample efficiency of our method compared to existing baselines. To our knowledge, this is the first work to study active exploration for online LLM alignment with fully online experimental verification. We hope our positive empirical results, along with the open-sourced codebase, will encourage future research in this direction, ultimately enabling LLMs to achieve superhuman intelligence with an affordable amount of human feedback.

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

# A  Algorithm details

While Algorithm 1 presents our Thompson sampling algorithm for LLM alignment, it is intractable and centered around the reward posterior modeling. We next present a practical sample-efficient alignment agent that learns both an LLM policy and an epistemic reward model (ERM) online.

---

**Algorithm 2** Sample-efficient alignment (SEA) for LLMs

**Input:** Reference policy $\pi_{\text{ref}}$, DAP loss function $F$, prompt distribution $p_{\mathcal{X}}$, unknown but queryable preference oracle $\mathbb{P}$, mixture ratio $\gamma$.

1: Initialize experience $\mathcal{D}_0 \leftarrow \varnothing$, policy $\pi_{\theta^0} \leftarrow \pi_{\text{ref}}$, and ERM weights $\Phi^0 = \{\phi_k^0\}_{k=1}^K$ randomly.
2: **for** $t = 1, \dots, T$ **do**
3:      Receive a prompt $\boldsymbol{x}_t \sim p_{\mathcal{X}}$.
4:      Sample $M$ responses $\boldsymbol{y}_t^i \sim \pi_{\theta^{t-1}}(\cdot|\boldsymbol{x}_t)$ to construct $\mathcal{S}_t = \{\boldsymbol{y}_t^i\}_{i=1}^M$.
5:      Sample $\phi \sim \text{Uniform}(\Phi^{t-1})$ and set $\boldsymbol{y}_t \leftarrow \arg\max_{\boldsymbol{b} \in \mathcal{S}_t} r_\phi(\boldsymbol{x}_t, \boldsymbol{b})$.        `// Select 1st response y.`
      `// E&E objective: aligning an online system.`
6:      **repeat**
         Sample $\phi \sim \text{Uniform}(\Phi^{t-1})$ and set $\boldsymbol{y}_t' \leftarrow \arg\max_{\boldsymbol{b} \in \mathcal{S}_t} r_\phi(\boldsymbol{x}_t, \boldsymbol{b})$.      `// Select 2nd response y'.`
         **until** $\boldsymbol{y}_t' \neq \boldsymbol{y}_t$
      `// BAI objective: labeling via crowdsourcing.`
7:      Set $\boldsymbol{y}_t' \leftarrow \arg\max_{\boldsymbol{b} \in \mathcal{S}_t} \mathbb{V}_\phi \left[\sigma\left(r_\phi(\boldsymbol{x}_t, \boldsymbol{y}_t) - r_\phi(\boldsymbol{x}_t, \boldsymbol{b})\right)\right]$,      `// OR select 2nd response y'.`
         where $\mathbb{V}_\phi[\cdot]$ computes variance across ensemble members of $\Phi^{t-1}$.
8:      **if** $g < \gamma$ for $g \sim \text{Uniform}(0, 1)$ **then**
         Label $\{\boldsymbol{y}_t, \boldsymbol{y}_t'\}$ with $\mathbb{P}$ to obtain $\mathcal{B}_t = \{(\boldsymbol{x}_t, \boldsymbol{y}_t^+, \boldsymbol{y}_t^-)\}$ and update experience $\mathcal{D}_t \leftarrow \mathcal{D}_{t-1} \bigcup \mathcal{B}_t$.
     **else**
         Use $\mathcal{R}_{\Phi^{t-1}}$ to get synthetic labels and obtain $\mathcal{B}_t = \{(\boldsymbol{x}_i, \tilde{\boldsymbol{y}}_i^+, \tilde{\boldsymbol{y}}_i^-)\}$.
     **end if**
9:      Update ERM with the regularized NLL loss (Eq. (8)):

$$\Phi^t \leftarrow \Phi^{t-1} - \alpha_\mathcal{R} \nabla_\Phi \mathcal{L}_\mathcal{R}(\Phi^{t-1}|\mathcal{D}_t).$$

                                                          `// Reward learning.`
10:     Update policy with the direct optimizer (Eq. (9)):

$$\theta^t \leftarrow \theta^{t-1} - \alpha_\pi \nabla_\theta \mathcal{L}_\pi(\theta^{t-1}|\mathcal{B}_t, \pi_{\text{ref}}, F).$$

                                                             `// Policy learning.`
11: **end for**

---

In Algorithm 2, we describe an online setting where a single example is processed at each time $t$ (batch size $b = 1$). This is mainly for notational convenience, while in implementation we set $b$ to be the training batch size (e.g., $128$). We instantiate the reward posterior with an epistemic reward model, which allows for efficient incremental update and sampling. We also replace the global optimization ($\arg\max_{\boldsymbol{b} \in \mathcal{Y}}$) with a policy-guided local search among proposals sampled from the latest online policy $\pi_{\theta^{t-1}}$. At each time $t$, we update ERM weights $\Phi$ with $m$ gradient steps with randomly sampled batches from the experience $\mathcal{D}_t$. We find setting $m = 5$ suffices to achieve a reasonable accuracy. The policy parameters $\theta$ are updated using mixed preference data, with a $\gamma$ proportion being the real environment experience and the remaining $(1 - \gamma)$ from the ERM's synthetic experience. Note that the synthetic experience is not added into $\mathcal{D}_t$ to ensure reward learning always uses ground truth environment data.

We consider the following three direct optimizers in our experiments:

- DPO [55]:

$$F_\theta(\boldsymbol{x}, \boldsymbol{y}^+, \boldsymbol{y}^-, \pi_{\text{ref}}) = -\log\sigma\left(\beta\log\frac{\pi_\theta(\boldsymbol{y}^+|\boldsymbol{x})\,\pi_{\text{ref}}(\boldsymbol{y}^-|\boldsymbol{x})}{\pi_{\text{ref}}(\boldsymbol{y}^+|\boldsymbol{x})\,\pi_\theta(\boldsymbol{y}^-|\boldsymbol{x})}\right) \tag{10}$$

- IPO [4]:

$$F_\theta(\boldsymbol{x}, \boldsymbol{y}^+, \boldsymbol{y}^-, \pi_{\text{ref}}) = \left(\log\left(\frac{\pi_\theta(\boldsymbol{y}^+|\boldsymbol{x})\,\pi_{\text{ref}}(\boldsymbol{y}^-|\boldsymbol{x})}{\pi_{\text{ref}}(\boldsymbol{y}^+|\boldsymbol{x})\,\pi_\theta(\boldsymbol{y}^-|\boldsymbol{x})}\right) - \frac{1}{2\beta}\right)^2 \tag{11}$$

- SLiC [88]:

$$F_\theta(\boldsymbol{x}, \boldsymbol{y}^+, \boldsymbol{y}^-, \pi_{\text{ref}}) = \max\left(0, 1 - \beta\log\frac{\pi_\theta(\boldsymbol{y}^+|\boldsymbol{x})\,\pi_{\text{ref}}(\boldsymbol{y}^-|\boldsymbol{x})}{\pi_{\text{ref}}(\boldsymbol{y}^+|\boldsymbol{x})\,\pi_\theta(\boldsymbol{y}^-|\boldsymbol{x})}\right) \tag{12}$$

 where $\beta$ controls the rate of deviation of $\pi_\theta$ from $\pi_{\mathrm{ref}}$.

# B  Full related works

**RLHF and DAP**. Commonly adopted RLHF pipelines [15, 65, 5, 50] first learn a proxy RM with a negative log-likelihood loss:

$$\mathcal{L}_r(\phi|\mathcal{D}) = -\mathbb{E}_{(\boldsymbol{x},\boldsymbol{y}^+,\boldsymbol{y}^-)\sim p_\mathcal{D}} \left[\log \sigma \left(r_\phi\left(\boldsymbol{x},\boldsymbol{y}^+\right) - r_\phi\left(\boldsymbol{x},\boldsymbol{y}^-\right)\right)\right], \tag{13}$$

where $\mathcal{D}$ is collected by querying human annotators using a behavior policy $\pi_{\mathrm{ref}}$ (typically the supervised fine-tuned policy $\pi_{\mathrm{sft}}$). Afterwards, *offline RL*[5] [36, 37] is conducted to learn $\pi_\theta$ with respect to the learned reward $r_\phi$ internally within the agent (Fig. 2a). However, the learned model $\pi_\theta$ might be inaccurate at regions out of the distribution (o.o.d.) of $\pi_{\mathrm{ref}}$ because little training data can be collected. An effective remedy is to incorporate a pessimistic term to combat the distributional shift, leading to a reformulation of the von Neumann winner policy objective in Eq. (4) as

$$J(\pi_\theta) = \mathbb{E}_{\boldsymbol{x}\sim p_\mathcal{X}} \mathbb{E}_{\boldsymbol{y}\sim \pi_\theta(\cdot|\boldsymbol{x})} \left[\underbrace{r_\phi(\boldsymbol{x},\boldsymbol{y})}_{\text{estimated } r^\star} \underbrace{- \beta \log \frac{\pi_\theta(\boldsymbol{y}|\boldsymbol{x})}{\pi_{\mathrm{ref}}(\boldsymbol{y}|\boldsymbol{x})}}_{\text{o.o.d. reward penalty}}\right] \tag{14}$$

$$= \mathbb{E}_{\boldsymbol{x}\sim p_\mathcal{X}} \left[\mathbb{E}_{\boldsymbol{y}\sim \pi_\theta(\cdot|\boldsymbol{x})} [r_\phi(\boldsymbol{x},\boldsymbol{y})] - \beta D_{\mathrm{KL}}(\pi_\theta(\cdot|\boldsymbol{x})||\pi_{\mathrm{ref}}(\cdot|\boldsymbol{x}))\right] \tag{15}$$

which converts an online objective regarding the human's implicit reward $r^\star$ to an offline objective regarding the proxy reward $r_\phi$. The KL penalty in Eq. (15) is widely used for language model fine-tuning [29, 80], and PPO [64] has become a default *RL optimizer* to maximize the KL-regularized reward. However, the performance of RLHF is guaranteed only if the preference data $\mathcal{D}$ induced by $\pi_{\mathrm{ref}}$ adequately covers $\pi^\star$ [90], which is often approximated by updating $\pi_{\mathrm{ref}}$ with the latest (improved) $\pi_\theta$ for re-sampling a batch of online experience and repeating Eq. (13) and (15). Prior works typically focus on offline or iterative online (with only a few iterations) settings [80, 18], which may compromise sample efficiency (Property 1).

True online RLHF is difficult due to the complexity and instability of RL optimizers. For example, Huang et al. [27] openly reproduces offline RLHF scaling behaviors but requires many implementation tricks for training, highlighting the difficulties of an online counterpart. Fortunately, the introduction of DAP (or *direct optimizers*) largely simplifies and stabilizes fine-tuning by conducting contrastive supervised learning directly on $\mathcal{D}$ (Fig. 2b). While most DAP works focus on learning from a fixed offline preference dataset (, including Zhao et al. [88], Rafailov et al. [55], Azar et al. [4], Meng et al. [46], Zhang et al. [87]), iterative DPO [81] observes improved results when allowing iterative online interaction. Guo et al. [24] further propose OAIF to make DAP faithfully online, satisfying Property 1, and demonstrate that online learning prevents over-fitting and yields continual performance improvement. Nevertheless, it still employs passive exploration strategies (using $\boldsymbol{y}, \boldsymbol{y}' \sim \pi_\theta$), hindering sample efficiency (Property 2).

**Online exploration in LLMs**. A line of recent works [44, 17, 45, 21] adopts the fully online bandit formulation and incorporates *active exploration* with uncertainty-aware RMs for response selection (Fig. 2c). In particular, Mehta et al. [44] consider the E&E setting and develop a UCB-style [3] algorithm; Das et al. [17] instead select the dueling responses with the most uncertain preference estimate, targeting the BAI setting in a pure exploration way; unlike the above, Melo et al. [45] view the problem from the angle of pool-based active learning and propose an acquisition function based on both entropy and epistemic uncertainty; finally, the work by Dwaracherla et al. [21] is the closest to ours in the sense that they apply double Thompson sampling (DTS) [78] for exploration, but DTS is designed for the E&E setting while they evaluate anytime average performance as in the BAI setting. We will show in App. G.1 that pure exploration by Das et al. [17] is not the best choice for BAI, and the objective mismatch in Dwaracherla et al. [21] could lead to suboptimal performance in respective settings. Meanwhile, all these works primarily focus on learning uncertainty-aware RMs online without updating LLM policies. Therefore, all responses are sampled from a fixed proposal policy $\pi_\beta$ (or even a fixed dataset), making the data coverage a critical concern.

Another line of research updates LLMs online while incorporating exploration. Zhang et al. [86] and Xie et al. [79] independently propose to learn an optimistic RM to encourage exploration. They

---

[5]*Offline* in the sense that $\pi_\theta$ is not directly learned from online human feedback. See App. C for details.

leverage the property of DPO [55] to reparameterize RM with policy and conclude with an extra optimistic term in the DPO loss function. Thus, their learning processes are like Fig. 2b but with an optimistic direct optimizer. Muldrew et al. [47] adopt the vanilla DPO loss but utilize the implicit reward margin to actively select dueling responses. Yet, these methods are tightly coupled with DPO and not compatible to other direct optimizers. Their experiments are also limited to a few online iterations, possibly due to the implementation difficulty of a faithfully online learning system. Given their relevance to our approach, we reproduce them in a fully online manner for fair comparisons in Sec. 6.1. We summarize prior works in Table 2.

|  | Method | Exploration | | Interaction | | | Proposal Policy | |
|---|---|---|---|---|---|---|---|---|
|  |  | Active | Passive | Online | Iterative | Offline | $\pi_\theta$ | $\pi_\beta$ |
| RL Optimizer | [15] |  | ✓ |  | ✓ | ✓ | ✓ |  |
|  | [65] |  | ✓ |  | ✓ | ✓ | ✓ |  |
|  | [5] |  | ✓ |  | ✓ | ✓ | ✓ |  |
|  | [50] |  | ✓ |  | ✓ | ✓ | ✓ |  |
| Direct Optimizer | [88] |  | ✓ |  |  | ✓ | ✓ |  |
|  | [55] |  | ✓ |  |  | ✓ | ✓ |  |
|  | [4] |  | ✓ |  |  | ✓ | ✓ |  |
|  | [46] |  | ✓ |  |  | ✓ | ✓ |  |
|  | [81] |  | ✓ |  | ✓ |  | ✓ |  |
|  | [24] |  | ✓ | ✓ |  |  | ✓ |  |
|  | [44] | ✓ |  | ✓ |  |  |  | ✓ |
|  | [17] | ✓ |  | ✓ |  |  |  | ✓ |
|  | [45] | ✓ |  | ✓ |  |  |  | ✓ |
|  | [21] | ✓ |  | ✓ |  |  |  | ✓ |
|  | [86] | ✓ |  | ✓ |  |  | ✓ |  |
|  | [79] | ✓ |  | ✓ |  |  | ✓ |  |
|  | [47] | ✓ |  | ✓ |  |  | ✓ |  |

**Table 2:** A summary of prior work. $\pi_\theta$ denotes the proposal policy that is continuously updated based on newly collected preference data, while $\pi_\beta$ denotes a fixed proposal policy. Algorithms that encompass online interaction (Property 1), active exploration (Property 2), and learnable $\pi_\theta$ offer the best sample efficiency. Notably, only three methods (listed at the bottom of the table) satisfy these characteristics, and we include them for comparisons in our experiments.

# C On connections with single-step RL

By viewing contextual dueling bandits as *single-step* preference-based RL (PbRL) [11, 77] problems, we can interpret paradigms shown in Fig. 2 from the RL perspective.

RLHF approaches (Fig. 2a) are instances of **offline model-based RL** [32, 84, 63, 41, 69], where they learn a reward model (no need for a transition model since the prompt-response interaction is single-step) of the environment from a batch of offline collected data, and train a policy (i.e., LLM) to maximize the return (i.e., expected one-step reward) with respect to the *learned* reward.

In contrast, DAP methods (Fig. 2b) are similar to **policy-based model-free RL** algorithms, e.g., REINFORCE [76] which conducts policy gradient update:

$$\mathbb{E}_{\boldsymbol{x}\sim\mathcal{X}}\mathbb{E}_{\boldsymbol{y}\sim\pi_\theta(\cdot|\boldsymbol{x})}\left[R(\boldsymbol{x},\boldsymbol{y})\nabla_\theta\log\pi_\theta(\boldsymbol{y}|\boldsymbol{x})\right], \tag{16}$$

where $R(\boldsymbol{x},\boldsymbol{y})$ is the return (i.e., cumulative reward) of the trajectory. To connect with DAP, we could set $R$ as arbitrary scalar values based on the binary preference outcomes, e.g., $R(\boldsymbol{x},\boldsymbol{y}^+)=\zeta$ and $R(\boldsymbol{x},\boldsymbol{y}^-)=-\zeta$ for preference triplet $\{\boldsymbol{x},\boldsymbol{y}^+,\boldsymbol{y}^-\}$. In this way we could rewrite Eq. (16) as

$$\mathbb{E}_{\boldsymbol{x}\sim\mathcal{X}}\mathbb{E}_{\boldsymbol{y},\boldsymbol{y}'\sim\pi_\theta(\cdot|\boldsymbol{x})}\mathbb{E}_{(\boldsymbol{y}^+\succ\boldsymbol{y}^-)\sim\mathbb{P}}\left[\zeta\left(\nabla_\theta\log\pi_\theta(\boldsymbol{y}^+|\boldsymbol{x})-\nabla_\theta\log\pi_\theta(\boldsymbol{y}^-|\boldsymbol{x})\right)\right], \tag{17}$$

by repeating action sampling twice and querying the oracle for preference labeling. This matches the gradient direction of contrastive DAP losses (e.g., see Section 4 of DPO [55]) if we optimize them online [24].

Additionally, active reward learning from behavior policy's data distribution (Fig. 2c) can be regarded as **inverse RL** [49], which tries to recover environment's reward function given expert trajectories. In the context of LLM alignment, the preference data $\{\boldsymbol{x},\boldsymbol{y}^+,\boldsymbol{y}^-\}_{i=1}^N$ directly encodes human's implicit reward $r^\star$, which can be inversely learned with assumptions such as the BT model [8]. However, existing methods belonging to this paradigm mostly rely on a fixed (and suboptimal) behavior policy for response sampling, whose coverage inherently limits the quality of the recovered reward function.

Last but not least, **SEA** depicted in Fig. 2d resembles a class of **online model-based RL** algorithms, known as Dyna [66, 28], that learns a *world model* from environment experience and trains a base agent (consisting of reactive policies and value functions) from both environment experience and model experience. Compared to model-free methods, Dyna naturally enables more sample-efficient learning by planning with the learned world model to update the base agent. In **SEA**, we learn the reward model online and update the LLM (i.e., the reactive policy) with model-planing experience by mixed preference learning (Sec. 4.2.3). Online model-based RL algorithms could suffer from catastrophic forgetting in the face of nonstationary data [42], and we leave it for future work. Overall, this model-based RL formulation is powerful and explains popular LLM techniques, e.g., Best-of-N sampling [72] can be viewed as planning for acting, which trades compute for performance. We believe it is a promising path leading us to unlock superhuman capabilities of LLMs.

## D Distributed learning framework

The interactive nature of LLM alignment necessitates an integrated online learning system that simulates the interface. The absence of a performant open-source online alignment system has restricted many existing works to only a few iterations of batch learning [47, 18, 14, 86, 79], which creates a mismatch with their theories that typically require a large number of online interaction rounds. Even worse, such absence also makes the comparison between different LLM exploration methods difficult, often restricting evaluations to the simplest iterative DAP baselines [86, 79].

To fill this gap, we build a highly efficient learning system for experimenting with online LLM alignment algorithms. We notice that the computational bottleneck lies in online response sampling (i.e., autoregressive generation) and preference labeling (e.g., human, large RMs, or large LLMs), which mirrors the slow actor-environment interaction seen in RL systems. Inspired by distributed deep RL systems which spawn many actors or environments in parallel [22, 75], we design an Actor-Learner-Oracle architecture for online LLM alignment, which is depicted in Fig. 5. The three types of workloads (i.e., actor, learner and oracle) are heterogeneous and require different optimization. In particular, we adopt vLLM [33] for the actor to accelerate the autoregressive response generation. We also use DeepSpeed's ZeRO [58, 57] strategies to enhance the memory efficiency of the learner. The updated model weights are broadcasted from the learner master to all actors after every optimizer

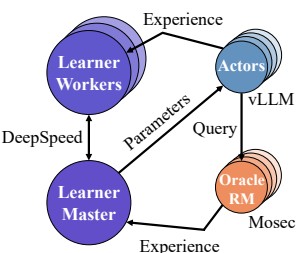

**Figure 5:** The learning system for experimenting online LLM alignment algorithms.

step efficiently via NCCL, similar to Hu et al. [26]. Furthermore, to improve the scalability, we wrap the oracle RM as a service using Mosec [83], which supports dynamic batching and parallel processing, to minimize preference query latency. Finally, we leverage DeepMind Launchpad [82] to compose all workloads into a distributed program and adopt Plasma [53] to efficiently transfer data across process boundaries.

We benchmark our system's efficiency against a concurrent implementation of online DPO by HuggingFace[6], which utilizes only DeepSpeed for memory optimization. Our system achieves up to **2.5**× latency reduction compared to this counterpart, demonstrating its computational efficiency. Due to space constraints, detailed benchmarking methods and results are presented in App. H.

## E Baseline methods

We review four baseline methods that are relevant to this work and used for comparisons in our experiments.

**Offline DAP**. We review DPO [55], which is a representative work in the direction of Direct Alignment from Preferences (DAP). It simplifies the two-stage pipeline of offline RLHF as a single step of supervised learning by leveraging the closed-form solution [52, 51] of the RL objective in Eq. (15):

$$\pi_r(\boldsymbol{y}|\boldsymbol{x}) = \frac{1}{Z(\boldsymbol{x})}\pi_{\text{ref}}(\boldsymbol{y}|\boldsymbol{x})\exp(\frac{1}{\beta}r(\boldsymbol{x},\boldsymbol{y})), \tag{18}$$

---

[6]https://huggingface.co/docs/trl/main/en/online_dpo_trainer.

where $Z(\boldsymbol{x})$ normalizes such that $\Sigma_{\boldsymbol{y}} \pi_r(\boldsymbol{y}|\boldsymbol{x}) = 1$, to reparametrize $r$ as a function of $\pi$:

$$r(\boldsymbol{x}, \boldsymbol{y}) = \beta \log \frac{\pi_r(\boldsymbol{y}|\boldsymbol{x})}{\pi_{\text{ref}}(\boldsymbol{y}|\boldsymbol{x})} + \beta \log Z(\boldsymbol{x}). \tag{19}$$

Consequently, plugging Eq. (19) into the reward model loss (Eq. (13)) yields a contrastive loss that directly optimizes the policy:

$$\min_{\pi_\theta} \mathbb{E}_{(\boldsymbol{x}, \boldsymbol{y}^+, \boldsymbol{y}^-) \sim p_{\mathcal{D}}} \left[ -\log \sigma \left( \beta \log \frac{\pi_\theta(\boldsymbol{y}^+|\boldsymbol{x}) \, \pi_{\text{ref}}(\boldsymbol{y}^-|\boldsymbol{x})}{\pi_{\text{ref}}(\boldsymbol{y}^+|\boldsymbol{x}) \, \pi_\theta(\boldsymbol{y}^-|\boldsymbol{x})} \right) \right], \tag{20}$$

where $\mathcal{D}$ is a pre-collected offline preference dataset.

We also experiment different DAP methods[7] besides DPO, such as IPO [4] and SLiC [88], whose loss functions are shown in Eq. (11) and (12).

**Online DAP** [24]. In contrast to the conventional DAP methods that learn a policy from a fixed dataset $\mathcal{D}$, online DAP proposes to collect on-policy preference data to update the policy online. It first samples responses from the current policy $(\boldsymbol{y}, \boldsymbol{y}') \sim \pi_{\theta_t}$, then acquires preference labels to form a batch $\mathcal{B}_t = \{(\boldsymbol{x}, \boldsymbol{y}^+, \boldsymbol{y}^-)\}_{i=1}^b$. One gradient step minimizing the DAP loss over this data batch to get $\pi_{\theta_{t+1}}$, which is used for the next iteration. Such approach not only mitigates the over-fitting issue faced by offline DAP methods [24], but also facilitates online interaction (Property 1) with the environment, falling into the second paradigm of CDB solution algorithms (Fig. 2b).

**Active Preference Learning (APL)** [47]. APL follows the online DAP paradigm, but is restricted to DPO due to its reliance on DPO implicit rewards. Two techniques are proposed by APL to actively select both prompts and dueling responses for querying the preference oracle:

1. *Predictive entropy (PE)* for selecting prompts. In this step APL computes a Monte-Carlo estimate of PE for each prompt as $\mathcal{H}_{\pi_\theta}(\boldsymbol{y}|\boldsymbol{x}) \approx -\Sigma_{n=1}^N \log \pi_\theta(\boldsymbol{y}_n|\boldsymbol{x})/N$, where $\boldsymbol{y}_n \sim \pi_\theta(\cdot|\boldsymbol{x})$ and $\log \pi_\theta(\boldsymbol{y}_n|\boldsymbol{x})$ is the summation of log probabilities of each token. Then, APL filters a subset of prompts with high PE to form $\mathcal{X}_S$.

2. *Preference model certainty* for selecting dueling responses. For prompts in $\mathcal{X}_S$, APL generates many responses for each prompt, then selects the pair with largest reward margin measured as $|\hat{r}(\boldsymbol{x}_i, \boldsymbol{y}_i) - \hat{r}(\boldsymbol{x}_i, \boldsymbol{y}_i')|$, where $\hat{r}$ is the DPO implicit reward $\hat{r}(\boldsymbol{x}, \boldsymbol{y}) = \beta(\log \pi_\theta(\boldsymbol{y}|\boldsymbol{x}) - \log \pi_{\text{ref}}(\boldsymbol{y}|\boldsymbol{x}))$.

By above two steps, APL actively explores more uncertain prompts and responses in an online DPO paradigm, satisfying both Properties 1 and 2.

**Exploratory Preference Optimization (XPO)** [79]. XPO studies LLM alignment in the framework of token-level MDP, and leverages the property that DPO conducts *implicit $Q^\star$-approximation* [56], so that

$$\beta \log \frac{\pi^\star(\boldsymbol{y}|\boldsymbol{x})}{\pi_{\text{ref}}(\boldsymbol{y}|\boldsymbol{x})} = r^\star(\boldsymbol{x}, \boldsymbol{y}) - V^\star(\boldsymbol{x}) \quad \forall \boldsymbol{y}, \tag{21}$$

where $V^\star$ is the optimal value function depending only on the prompt $\boldsymbol{x}$. XPO incorporates the *implicit (global) optimism* for exploration by overestimating the value $V_{\pi_\theta}(\boldsymbol{x}) = r^\star(\boldsymbol{x}, \boldsymbol{y}) - \beta \log \frac{\pi_\theta(\boldsymbol{y}|\boldsymbol{x})}{\pi_{\text{ref}}(\boldsymbol{y}|\boldsymbol{x})}$. This is achieved by optimizing the policy with a modified DPO loss:

$$\min_{\pi_\theta} \mathbb{E}_{(\boldsymbol{x}, \boldsymbol{y}^+, \boldsymbol{y}^-, \boldsymbol{y}^{\text{ref}}) \sim p_{\mathcal{B}^t}} \left[ \alpha \log \pi_\theta(\boldsymbol{y}^{\text{ref}}|\boldsymbol{x}) - \log \sigma \left( \beta \log \frac{\pi_\theta(\boldsymbol{y}^+|\boldsymbol{x}) \, \pi_{\text{ref}}(\boldsymbol{y}^-|\boldsymbol{x})}{\pi_{\text{ref}}(\boldsymbol{y}^+|\boldsymbol{x}) \, \pi_\theta(\boldsymbol{y}^-|\boldsymbol{x})} \right) \right], \tag{22}$$

where $\boldsymbol{y}^{\text{ref}} \sim \pi_{\text{ref}}(\cdot|\boldsymbol{x})$ and $\mathcal{B}^t$ is an on-policy data batch in the same vein as online DPO. Intuitively, the first term in Eq. (22) biases the policy toward a large value estimation such that $V_{\pi_\theta} \gtrsim V^\star$, implementing the optimism in the face of uncertainty (OFU) for exploration. Theoretically, Xie et al. [79] also prove the sample complexity bound of XPO, making it a promising algorithm for online LLM alignment.

Self-exploring language model (SELM) [86] is a concurrent work of Xie et al. [79] that proposes nearly the same theoretic algorithm to achieve OFU. However, the practical implementation of SELM involves offline preference dataset for training, making it hard to benchmark in an online alignment setting like ours. Therefore, we will keep XPO as our baseline for comparison.

---

[7]We use "DAP method" and "direct optimizer" interchangeably.

# F  Full experimental details

In the main text we focus on the task of summarization using the TL;DR dataset. This provides a lightweight and clean setting to extensively study different algorithmic designs with affordable computational resources. App. F.1 provides the full details of this setting.

To further validate the sample efficiency of **SEA** in aligning LLMs to perform general tasks, we adopt the UltraFeedback dataset [16] and evaluate trained LLMs on AlpacaEval 2.0 [38]. App. F.2 provides more details of this setting.

## F.1  Details of TL;DR task

**Models**. We experiment three model scales (1B, 2.8B, 6.9B) from the Pythia family [7]. We take pretrained SFT models from [27] as $\pi_{\mathrm{ref}}$ for the starting model in all experiments. Except in Sec. 6.1, we use 1B model for other experiments to save computation.

**Preference oracle**. We simulate the process of human feedback with a strong scalar RM and refer it as preference oracle. We choose Skywork-Reward-Llama-3.1-8B[8] [40], which is top-ranked in RewardBench leaderboard [35], as the preference oracle.

**Epistemic reward model**. We build ERM on top of a pretrained 0.4B transformer [30], by removing its head and adding an ensemble of MLPs. The size of ensemble is set to $K = 20$, and all MLPs contain 2 hidden layers of 128 nodes. Note that the ERM is chosen to be much smaller than the preference oracle following Dwaracherla et al. [21], which reflects the fact that human preferences can be more complex than what the agent can model. The regularization coefficient $\lambda$ is fixed to be $0.5$ after a coarse hyperparameter search.

**Data**. We employ the widely adopted TL;DR dataset [65] for our experiments. It consists of Reddit posts as prompts, and the agent is required to give summaries that align with human preferences. We fix 50k prompts for training and limit the query budget to 50k as well.

**DAP methods**. We adopt three DAP methods (direct optimizers) to thoroughly validate our algorithm, including DPO [55], IPO [4] and SLiC [88]. Except in Sec. 6.1, all experiments are done with DPO as the direct optimizer.

**Baselines**. Similar to Guo et al. [24], we include the offline and online variants of different DAP methods as baselines. Additionally, we compare with two active exploration baselines built on online DPO: APL [47] and XPO [79]. A detailed review of all baselines can be found in App. E.

**Metrics**. We use the win rate of agent's responses against reference responses judged by the preference oracle as the performance metric. This metric can reflect both the agent's cumulative regret and anytime regret (i.e., average performance). In the E&E setting, we measure the "online" win rate of the agent's dueling responses that are executed during experience collection and take the average. In the BAI setting, we measure the "offline" win rate by evaluating the latest agent's responses given a fixed set of 1000 holdout prompts periodically. We mainly focus on the BAI setting because crowdsourcing seems a major scenario for most practitioners, and present one set of experiments for comparing different exploration strategies in both settings. When the comparison is only made within a model scale, we report the relative win rate against the initial STF models. When the comparison is across scales (Fig. 1 Left), we report the absolute win rate against the ground truth responses in the dataset.

**Hyperparameters**. We set $\beta = 0.1$ for DPO and $\beta = 0.2$ for SLiC and find they are robust for all scales. We tune $\beta$ from $\{0.2, 0.3, 0.5, 1.0\}$ for IPO across scales and report the best performing results. We sample $M = 20$ on-policy responses with a temperature $\eta = 0.7$ during training, and use greedy decoding for offline evaluation (BAI's metric). We use the Adam optimizer with learning rate of $5 \times 10^{-7}$ and cosine scheduling, and set the batch size to 128. We initialize the mixture ratio $\gamma$ of **SEA** to be 1 and adjust it to $0.7$ after a burn-in period of 1k samples.
All hyperparameters are kept the same for offline and online baselines, except that online methods update the sampling policy after every gradient step as the latest $\pi_{\theta_t}$. For APL and XPO, we keep the learning rate and DPO's $\beta$ the same for apple-to-apple comparisons. Specifically for APL, we initially sample 1024 prompts per batch and use the predictive entropy to filter a subset of 128 prompts. Then, we sample 8 responses per prompt and use the preference model certainty to finalize two responses

---

[8]https://huggingface.co/Skywork/Skywork-Reward-Llama-3.1-8B.

for the duel. Specifically for XPO, we follow the their recommended optimism coefficient to set $\alpha = 5 \times 10^{-6}$.

**Statistical significance**. There are various factors to introduce randomness during online learning. We thus launch 3 independent runs for every experiment with different random seeds. All the results are reported with mean and standard error to indicate their statistical significance.

**Computational resources**. Experiments at all scales are conducted on a single machine with 8 A100 GPUs to run the learner and actors. We additionally host a separate remote server with workers spawned on 16 A100 GPUs for the oracle RM[9], so that it can be queried by all concurrently running experiments. All experiments conducted for this research consume about 2 A100 GPU years.

### F.2  Details of general tasks

**Model**. Following Meng et al. [46], Zhang et al. [86], we employ `Llama3-8B-Instruct`[10] as our initial model $\pi_{\text{ref}}$.

**Preference oracle**. We follow Meng et al. [46] to adopt `ArmoRM-Llama3-8B-v0.1`[11] [74] as the preference oracle to provide online preference feedback.

**Data**. We take the UltraFeedback dataset [16], which is widely used for LLM alignment in the literature. We filter out samples whose prompt is longer than 1800 tokens and result in 61k samples. We extract prompts from the filtered dataset while excluding the responses. The prompt set are collected from multiple sources and cover diverse domains, making it suitable to improve LLM's capability on general tasks.

**DAP method and baselines**. We employ the state-of-the-art DAP method, SimPO [46], as our direct optimizer. Since SimPO is originally an offline algorithm, we extend it to Online SimPO and take both offline and online variants as baselines.

**Evaluation**. We evaluate **SEA** and baselines using AlpacaEval 2.0 [38]. It consists of 805 test prompts, and uses GPT4-Turbo to judge the quality of model responses against reference responses generated by `GPT-4-Turbo`. We follow the standard protocol to report both the win rate (WR) and the Length-Controlled win rate (LC) [19].

**Hyperparameters**. We follow SimPO's recommended hyperparameters to set $\beta = 10$ and $\gamma/\beta = 0.3$. We use a learning rate of $8 \times 10^{-7}$ and batch size of 128. The decoding temperature is set to be 0.9 for generating evaluation outputs. The same hyperparameters apply to baselines and our method. Configurations of **SEA** are kept the same as those in the `TL;DR` task (App. F.1).

## G  Extended empirical studies

We present additional empirical studies in this section, including investigation on different exploration strategies (App. G.1) and preference oracles (App. G.2) on the `TL;DR` task, as well as the performance comparison on AlpacaEval 2.0 for general tasks (App. G.3).

**G.1  Choice of exploration strategies**

Recalling that different LLM alignment scenarios (online system or crowdsourcing) require different exploration strategies to meet their respective learning objectives (Sec. 2.2). We investigate three strategies based on posterior sampling and compare them on both online and offline performance. The first strategy (Uncertainty) focuses on pure exploration with information maximization. It seeks the pair of dueling responses that exhibits the largest epistemic uncertainty, which is implemented by selecting the pair whose logits difference has the largest variance across ensemble members. The second (E&E-TS) and the third (BAI-TS) strategies follow the principles in Algorithm 1, and their differences are between Line 5 and Line 6. The comparison results are shown in Fig. 6 (Left and Middle). Focusing on the left plot, we observe that E&E-TS strategy achieves the best online performance, which is within our expectation. In contrast, Uncertainty shows the worst online performance because it tries to maximize the information gain but does not prioritize reward maximization. On the other hand, conclusions are interestingly different when taking the offline performance as the metric. In this case, BAI-TS and

---

[9]We utilize the Kubernetes service for routing requests to multiple Mosec [83] instances.

[10]https://huggingface.co/meta-llama/Meta-Llama-3-8B-Instruct.

[11]https://huggingface.co/RLHFlow/ArmoRM-Llama3-8B-v0.1.

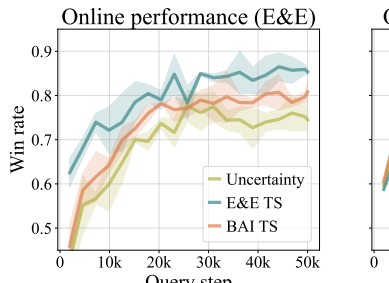
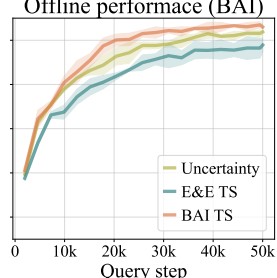
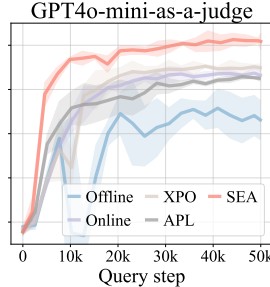

**Figure 6: (Left and Middle)** Win rate comparison of different exploration strategies measured in E&E and BAI settings. **(Right)** Win rate comparison of different agents when using `GPT4o-mini` to simulate human feedback via LLM-as-a-judge.

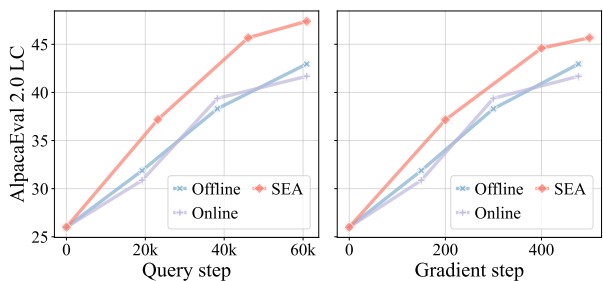

**Figure 7:** LC win rates on AlpacaEval 2.0 with respect to query budget and gradient update budget.

**Table 3:** AlpacaEval 2.0 results. LLM exploration methods are highlighted in blue.

| Model | LC | WR |
|---|---|---|
| GPT-4 Omni (05/13) | 57.5 | 51.3 |
| GPT-4 Turbo (04/09) | 55.0 | 46.1 |
| Yi-Large Preview | 51.9 | 57.5 |
| **SEA**+SimPO | **47.4** | **41.1** |
| Claude 3 Opus (02/29) | 40.5 | 26.1 |
| SELM | 34.7 | 34.8 |
| XPO | 29.4 | - |
| Llama 3 8B Instruct | 22.9 | 22.6 |

Uncertainty both exhibit more efficient offline performance improvement than E&E-TS. This can be attributed to that exploration for uncertainty minimizing helps to identify more informative responses to train the LLM policy. Moreover, BAI-TS ≻ Uncertainty indicates exploration with both reward and information maximization is better than exploration with only information maximization. E&E-TS, however, always chooses two responses with similarly high quality to exploit. This can not only lead to less efficient exploration, but also result in less efficient policy learning due to smaller DAP loss gradients.

### G.2 Aligning LLMs with a human simulator

Results presented so far are based on experimenting LLM alignment with the preference oracle being a scalar reward model, which is deterministic and does not capture the potential randomness of the choice by real humans. To test different agents in a more realistic setting, we use generative models as human simulator in an LLM-as-a-judge [10, 89] manner. In particular, we directly query the OpenAI API and use `gpt-4o-mini-2024-07-18` as the judge to provide preference feedback. We use a similar prompt template to Li et al. [38]'s, which is shown in Fig. 10. We also randomly swap the order of two responses to mitigate the known position bias of LLM judges. The results are shown in Fig. 6 (Right). We can observe the performance curves generally exhibit higher variance, possibly due to the randomness introduced in the feedback process, which puts more stringent requirements for learning algorithms. The two active exploration methods demonstrate opposite results to those in Sec. 6.1—APL learns fast initially but is eventually outperformed by Online, while XPO improves over Online after stabilizing its training and delivers a better final performance. Our agent, **SEA**, is shown to offer the best sample efficiency as well as asymptotic performance, further validating the importance of online learning and well-designed active exploration mechanism.

### G.3 Performance on general tasks

We investigate the generalizability of **SEA** by training with the prompt set from UltraFeedback [16] and evaluating the model performance on AlpacaEval 2.0 [38]. Fig. 7 shows the Length-Controlled (LC) win rate of different models against `GPT-4-Turbo`. The left plot compares the sample efficiency (in terms of the number of queries) of offline, online and **SEA** SimPO. The results suggest that enabling online interaction does not improve the sample efficiency over the offline counterpart. Such observation is in stark contrast to what we have seen in the `TL;DR` task, where the online agent always improves over the offline ones. We hypothesize that this is due to the different coverage of $\pi_{\text{ref}}$ in

these two tasks. For `TL;DR`, which is a much easier task, the initial SFT models already have good coverage, permitting online DAP with only *passive exploration* to work reasonably well; however, for more challenging tasks, the insufficient coverage of $\pi_{\text{ref}}$ would lead to sample complexity exponential in $\frac{1}{\beta}$ [79], which necessitates *deliberate exploration*, such as Thompson sampling proposed in this work. The above claim is justified by observing that **SEA** largely improves the sample efficiency over the online and offline variants.

Attentive readers may have noticed that comparing query budget could be advantageous to **SEA** because pseudo labels are used in mixed preference learning (Sec. 4.2.3), which results in more gradient steps given the same query budget. In the right plot of Fig. 7, we show the performance versus gradient step. We can observe **SEA** has the steepest learning curve, verifying that it explores more informative samples to yield faster improvement.

Last but not least, in Table 3, we show the AlpacaEval 2.0 LC win rates of XPO and SELM (as reported in their papers), along with ours and several cutting-edge LLMs. **SEA** is agnostic to direct optimizers, thus it can leverage the state-of-the-art SimPO to achieve a high LC of $47.4\%$. On the other hand, XPO and SELM can only be applied to DPO, restricting their potential to incorporate future advances in direct optimization algorithms.

# H  System benchmarking

We conduct a rigorous benchmarking comparison on the efficiency of online DPO training using our learning system, alongside the `trl`'s implementation[12].

**Settings**. In alignment with the examples provided by `trl`, we use the TL;DR [65] dataset and evaluate training efficiency at three model scales: 1B, 2.8B and 6.9B parameters for both SFT-ed LLMs[13] and exclusively trained RMs[14]. This is similar to the settings in our experiments (see App. F) except that we fix the preference oracle to be a strong general-purpose RM.

**Hardware & Software**. All benchmarking experiments are conducted on a single machine with eight A100-40G GPUs and 96 AMD EPYC 7352 CPUs. To ensure fair comparison, we align all key hyperparameters for both our codebase and `trl`. The DeepSpeed ZeRO-2 strategy is employed by default when GPU memory suffices; otherwise, ZeRO-3 or ZeRO-2-offload is utilized as applicable. Notably, the distributed architecture of our implementation provides flexibility in system configuration, enabling adjustments to accommodate memory and computational time constraints. Fig. 8 illustrates two example configurations employed in our benchmarking experiments. We will provide all benchmarking scripts in our codebase for reproducibility.

- **Config 1** collocates all three workloads on each of the GPUs. Specifically, eight vLLM instances (for actors) and eight Mosec workers (for oracle RMs) are spawned to run independently on each GPU. After a batch of responses is generated (by actors) and labeled (by oracle RMs), it is sent to the learner, which runs on all eight GPUs coordinated through ZeRO strategies for policy learning. The updated policy weights are then broadcasted to all actors for *on-policy* response sampling on subsequent prompt batch. While this configuration maximizes GPU utilization, it requires substantial GPU memory to accommodate all workloads and is thus employed only for 1B scale experiments.

- **Config 2** only collocates actor and oracle workloads on half of the GPUs, reserving the remaining four GPUs exclusively for the learner. This is suited for larger-scale experiments (e.g., 2.8B or 6.9B), where additional GPU memory is allocated to the learner. However, this setup incurs idle time on half of the GPUs due to data dependency, as the learner must await new preference data, and the actor must await updated policies. An alternative is to implement *asynchronous* data collection, where minor data staleness is allowed by using $\theta_{t-1}$ to generate data for updating $\theta_t$. Although this data would not be strictly on-policy,

---

[12]https://github.com/huggingface/trl/blob/main/trl/trainer/online_dpo_trainer.py.
[13]https://huggingface.co/trl-lib/pythia-1b-deduped-tldr-sft;https://huggingface.co/trl-lib/pythia-2.8b-deduped-tldr-sft;https://huggingface.co/trl-lib/pythia-6.9b-deduped-tldr-sft
[14]https://huggingface.co/trl-lib/pythia-1b-deduped-tldr-rm;https://huggingface.co/trl-lib/pythia-2.8b-deduped-tldr-rm;https://huggingface.co/trl-lib/pythia-6.9b-deduped-tldr-rm

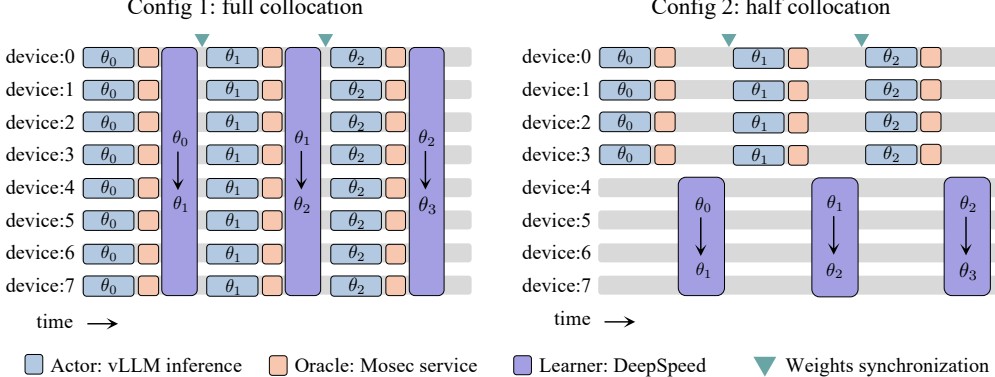

**Figure 8:** Two example configurations of our learning system used in benchmarking experiments.

asynchronous training could reduce idle time and enhance GPU utilization. This approach has proven effective in large-scale RL systems [6], and we leave this optimization to future work.

**Results**. Benchmarking results for the latency of training a batch of 128 samples are presented in Fig. 9. Overall, training with the config 2 demonstrates consistently greater efficiency than `trl`, achieving up to a **2.5×** reduction in latency at the 2.8B scale.

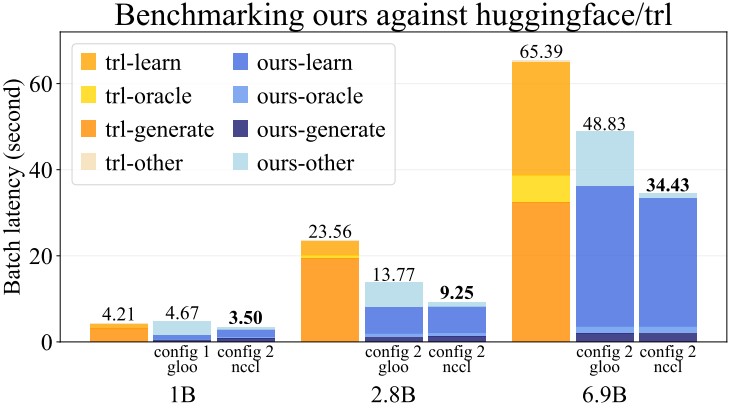

**Figure 9:** Averaged training latency (over 10 batches, equivalent to 1280 samples) comparing ours against `huggingface/trl`.

We next analyze the time costs for individual stages: generate, oracle and learn. Across all scales and configurations, ours demonstrates significantly lower *generate* time than `trl`, due to distributed actors utilizing vLLM. Additionally, at the 6.9B scale, ours requires substantially less *oracle* time than `trl`, as `trl` employs ZeRO-3 to prevent GPU memory overflow, thereby slowing inference. In contrast, ours config 2 allows for flexible collocation, enabling oracle RMs hosted via Mosec to operate in parallel without sharding. However, ours config 2 incurs longer *learn* time compared to `trl` due to the use of only half the available GPUs. This limitation also explains why, at the 1B scale, config 2 has higher latency than config 1 across all stages.

The *other* category accounts for time costs associated with data loading, tokenization, and communication. Here, inter-process communication is the primary cost, with `trl` showing minimal overhead as all three stages operate within the same process on identical micro-batches, avoiding weight synchronization. By contrast, ours requires considerable time to transfer updated policy weights from the learner to all actors. While NCCL is recommended for synchronization over GLOO, it requires older vLLM packages (prior to version 0.4.3), which may lack support for newer LLM architectures. Moreover, NCCL is incompatible with config 1 due to its restriction on the learner master process establishing two separate process groups (one for DeepSpeed, the other for weight synchronization). In summary, we recommend future researchers prioritize the config 2 and employ NCCL when feasible.

# I Additional materials

In this section, we include a comparison of prior works (Table 2), the prompt template we use for LLM-as-a-judge (Fig. 10), and an example showing ChatGPT actively explores (Fig. 6).

---

**LLM-as-a-Judge Prompt Template**

I require a leaderboard for various large language models. I'll provide you with prompts given to these models and their corresponding outputs. Your task is to assess these responses, and select the model that produces the best output from a human perspective.
## Instruction
{instruction}
## Model Outputs
Here are the unordered outputs from the models. Each output is associated with a specific model, identified by a unique model identifier.
{
"model_identifier: "0",
"output": {response0}
}
{
"model_identifier: "1",
"output": {response1}
}
## Task
Evaluate the models on the basis of the quality and relevance of their results, and select the model that generated the best result. Reply with the identifier of the best model. Our evaluation will only take into account the first character of your answer, so make sure it contains only one of the identifiers and nothing else (no quotation marks, no spaces, no new lines, ...).

---

**Figure 10:** We provide the prompt with two responses and ask GPT to choose the preferred one. We parse the first token's logits of "0" and "1" and apply the BT model to obtain the relative preference.

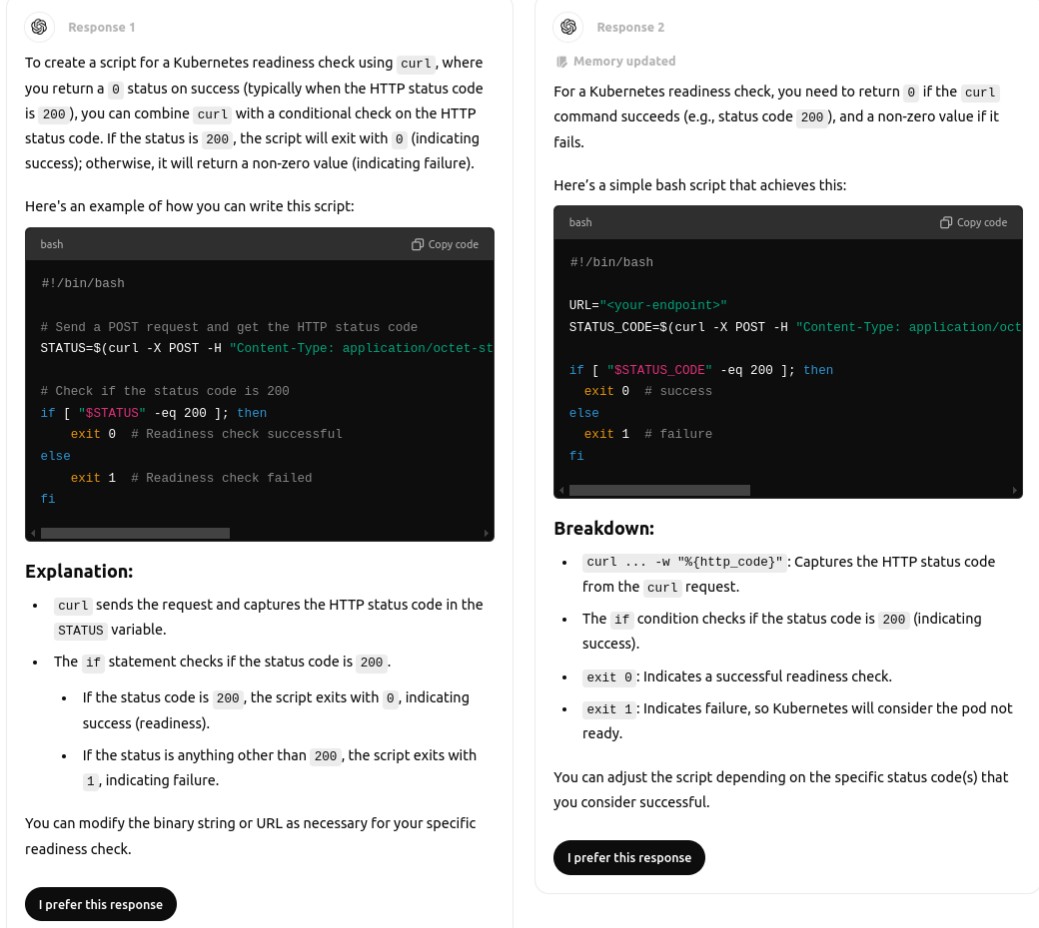

**Figure 11:** ChatGPT system asks for users' preference feedback to strategically explore better answers. In this case, algorithms should be designed around the objective of *minimizing cumulative regret* (i.e., the E&E setting), because the quality of both responses generated by the system affects user experience.

