# OpenReview forum: "Sample-Efficient Alignment for LLMs"
_NeurIPS.cc/2025/Conference — Submitted to NeurIPS 2025_

### Official Review · Reviewer_amjx · 2025-06-11

**Clarity:** 3
**Significance:** 3
**Originality:** 2
**Rating:** 4
**Confidence:** 3

**Summary:**

The paper proposes a novel approach to align large language models (LLMs) with human preferences using minimal human feedback. It formulates the alignment problem as a contextual dueling bandits (CDB) framework, where LLMs generate response pairs for given prompts, and human feedback provides pairwise preferences. The authors introduce the Sample-Efficient Alignment (SEA) algorithm, which leverages Thompson Sampling (TS) with deep ensembles to estimate reward uncertainty and guide exploration. SEA supports two alignment scenarios: Explore & Exploit (E&E) for real-time interaction and Best Arm Identification (BAI) for offline optimization. The algorithm uses a policy-guided search to manage the high-dimensional response space and incorporates mixed preference learning to balance human and synthetic feedback.

**Questions:**

1. A baseline, SELM, is deleted since `involves offline preference dataset' as claimed. However, why it can not be compared with offline BAI setting?

2. Can the authors explicitly clarify what aspects of SEA are novel and how they mechanistically differ from prior active exploration methods (e.g., APL, XPO) beyond empirical performance gains?

3. What is the difference between the 'policy-guided search and mixed preference learning techniques', and' sampling-based optimization in bandit literature and contrastive training and model-based RL concepts'?

**Ethical Concerns:**

["NO or VERY MINOR ethics concerns only"]

**Final Justification:**

The author has effectively addressed my concerns. Additionally, I have considered the perspectives of other reviewers.

**Limitations:**

1. The choice of a 0.4B reward model and Pythia family maybe outdated and potentially impractical.
2. The robustness of the SEA algorithm to noisy or inconsistent human feedback, which is common in real-world settings.

**Quality:**

3

**Strengths And Weaknesses:**

Strengths:

1. The contextual dueling bandits (CDB) framework provides a unified perspective for LLM alignment and the SEA algorithm is theoretically solid.
2. The application of TS with deep ensembles to LLM alignment is an adaptation, addressing computational challenges with policy-guided search and mixed preference learning.
3. The distinction between E&E and BAI scenarios is a unique contribution, clarifying different alignment needs.
4. The algorithm's flexibility is demonstrated by its compatibility with multiple direct optimizers (DPO, IPO, SLiC), enhancing its practical utility.

Weaknesses:

1. The core idea of framing LLM alignment as a contextual dueling bandit (CDB) builds heavily on prior bandit literature. While applying CDB to LLM alignment is novel, the paper does not sufficiently highlight what aspects of this framing are unique compared to existing RLHF or DAP methods. For example: a. The connection between RLHF/DAP and CDB is descriptive but does not introduce fundamentally new theoretical insights. b. The use of TS (Algorithm 1) is a direct adaptation of bandit algorithms, with modifications (e.g., ERM, policy-guided search) that are practical but not groundbreaking

2. The active exploration strategies in SEA (e.g., BAI-TS, E&E-TS) share similarities with prior methods like APL and XPO, which also aim to improve sample efficiency through exploration. The paper does not clearly articulate how SEA’s TS-based approach differs mechanistically from these methods, beyond empirical performance gains.

---

> ### Author Rebuttal · Authors · 2025-07-31
>
> Thank you for your insightful questions and valuable suggestions to add more experiments. We hope to address your concerns below.
>
> ***Q1: Why is SELM excluded?***
>
> Thank you for raising this point. From both the SELM paper and our personal communication with the authors, we understand that **SELM is theoretically equivalent to XPO**. Their release dates are also nearly simultaneous (May 29, 2024 for SELM vs. May 31, 2024 for XPO), and we consider them **concurrent works based on the same core idea**. This was our primary reason for not including SELM separately in our experiments, as noted in line 794.
>
> Additionally, we observed that **SELM's official implementation uses an offline preference dataset for training**, which diverges from both their proposed algorithm and the online alignment setting our work focuses on. We mention this discrepancy in the text for transparency, although it was not the reason for omitting SELM as a baseline.
>
> We also wish to clarify a potential misconception: training with an offline dataset, as in SELM’s implementation, **does not qualify as offline Best-Arm Identification (BAI)**. Both E&E (Exploration & Exploitation) and BAI assume an **online** interaction loop, where responses are generated and feedback is collected in real time. The distinction lies in evaluation:
> * E&E focuses on the quality of **every online decision**,
> * BAI evaluates the **final learned model** after online training.
>
> We will revise the text to make these clarifications more explicit.
>
> ***Q2.1 (Weakness 1): Novel aspects of SEA***
>
> The main novelty of this paper lies in clarifying E&E and BAI scenarios in the context of LLM alignment, proposing a unified Thompon sampling-based algorithm for both E&E and BAI, and providing novel solutions to the technical challenges when conducting TS in high dimensional space like LLM. These developments have practical implications that can more efficiently scale LLM alignment from human preferences. We also present extensive empirical evidence to validate the method's effectiveness with an extendable codebase that can benefit future research.
>
>
> We will revise the manuscript to make these contributions more explicit and better emphasize the technical and practical significance of SEA.
>
>
>
> ***Q2.2 (Weakness 2): How does SEA mechanistically differ from prior methods?***
>
> Although APL and XPO aim to improve sample efficiency through exploration, both are **tightly coupled to DPO as their loss function**. Specifically:
> * APL selects pairs with the largest implicit reward margin (from DPO) to guide exploration, relying entirely on the DPO reward proxy.
> * XPO adds an explicit optimism term to the DPO loss, effectively injecting a global exploration bias but still constrained within the DPO framework.
>
>
> In contrast, SEA introduces a loss-agnostic, algorithmic framework based on contextual dueling bandits and Thompson sampling (TS). Instead of relying on DPO-specific signals, SEA maintains a posterior over reward models and samples duels in a Bayesian manner to explore uncertainty regions more systematically.
>
> Key mechanistic differences include:
> * **Exploration principle**: SEA uses posterior sampling (TS), while APL relies on handcrafted scoring functions within DPO and XPO encourages global optimism (also within DPO).
> * **Policy optimizer independence**: SEA is compatible with any downstream optimizer (e.g., DPO, IPO, SLiC), while APL/XPO are tied to DPO.
>
> We believe this principled, modular approach enables broader applicability and better scalability in online alignment. We will revise the paper to highlight this distinction more explicitly.
>
>
> ***Q3: Difference between the 'policy-guided search and mixed preference learning techniques', and 'sampling-based optimization in bandit literature and contrastive training and model-based RL concepts'***
>
> Thanks for this insightful question. "Policy-guided search" is a computationally efficient way of getting the "best arm" in the large language response space, thus it's related to high-dimensional approximate sampling in the literature [1,2]. In a simple bandit, once the reward model is sampled, finding the best arm for the sampled reward model is computationally easy. This becomes computationally difficult in the LLM setting as the number of “arms” grows exponentially with sequence length motivating the use of sampling-based optimization. Our novelty lies not in this part, but in the “mixed preference learning” part where we propose a method for efficiently maintaining a sampler to sample good sequences for a mix of reward functions instead of sampling for a single reward function in sampling-based optimization.
> In "mixed preference learning", we mix in both the real preference data together with preferences labelled using all the different reward models that we maintain, when learning the LLM policy. The learned policy can hence be used to sample sequences that are good for all the different reward models rather than for a single reward model. The optimization is usually done with contrastive training (e.g., DPO). This is also related to model-based RL where the world model's simulated data can be used to learn the policy; here our world model contains the set of the different reward models that are still being explored.
>
> ***Limitations: 0.4B RM and Pythia may be outdated and the robustness of SEA under noisy feedback***
>
> Thank you for this valuable suggestion. To further validate the generality of SEA, we conducted additional experiments with the following new setting:
> * Base LLM: `meta-llama/Llama-3.2-1B-Instruct`
> * Prompt Dataset: `HuggingFaceH4/ultrafeedback_binarized` (with 60k/2k train/test splits)
> * Reward Oracle: `Skywork-Reward-V2-Llama-3.1-8B`
> * Reward Model Backbone: `Skywork/Skywork-Reward-V2-Llama-3.2-1B`
> * Preference noise: preference is flipped with 10% probability if enabled
> * Evaluation temperature: 0.6
>
> This setup uses a different model family (Llama 3) and a more realistic dataset (UltraFeedback). We also leverage a stronger reward oracle (Skywork-V2-8B) and a larger reward model backbone (Skywork-V2-1B).
>
> | step | sea-dpo | online-dpo | sea-dpo-noisy | online-dpo-noisy |
> |------|---------|------------|---------------|------------------|
> |    0 |    0.48 |       0.49 |          0.48 |             0.49 |
> |  100 |    0.59 |       0.51 |          0.54 |             0.50 |
> |  200 |    0.56 |       0.54 |          0.58 |             0.54 |
> |  300 |    0.58 |       0.58 |          0.60 |             0.57 |
> |  400 |    0.60 |       0.59 |          0.62 |             0.58 |
> |  500 |    0.63 |       0.61 |          0.63 |             0.59 |
>
> The table reports win rates against the initial model during training. SEA-DPO consistently learns faster and converges to a higher win rate than Online-DPO, reinforcing the effectiveness and generality of our approach across model families and datasets.
>
> In addition, we tested both SEA and online DPO under noisy preference feedback (with 10% chance the preference is flipped). As shown in the table, SEA is robust to feedback noise and shows superior learning efficiency and final performance over the baseline.
>
> ---
>
> Thank you again for your detailed feedback and suggestions, and we hope our responses and new experiments were able to address any remaining concerns. Please do let us know if you have any further questions as well as what would be expected for score improvement.
>
> ---
>
> [1] Mazumdar, Eric, et al. "On approximate Thompson sampling with Langevin algorithms." international conference on machine learning. PMLR, 2020.
>
> [2] Osband, Ian, et al. "Approximate thompson sampling via epistemic neural networks." Uncertainty in Artificial Intelligence. PMLR, 2023.

---

> > ### Comment · Reviewer_amjx · 2025-08-05
> >
> > Thank you for your response, which effectively addresses my concerns. I will upgrade my rating to a positive score. Additionally, I suggest exploring the use of bandit algorithms for LLM sample alignment. Have the authors considered employing Conversational Bandit approaches, such as ``Multi-Agent Conversational Online Learning, to achieve more efficient human preference alignment''?

---

> > > ### Author Response · Authors · 2025-08-05
> > > **Thank you!**
> > >
> > > Thank you for your constructive feedback and for raising our score!
> > >
> > > We find the idea of applying the Conversational Contextual Bandit (CCB) framework to LLM alignment particularly compelling. We concur that the core strength of this approach lies in its ability to transform the alignment process from a passive, single-turn interaction into an active, multi-turn dialogue. In this model, the LLM functions as a bandit agent that can strategically choose between asking a clarifying question (pulling a "key-term" arm) and providing a final answer (pulling a "response" arm) [1].
> > >
> > > By actively seeking feedback, the LLM can gather more nuanced and specific information, which we believe could lead to more efficient and robust preference alignment. The challenge of designing and training a multi-agent system to effectively simulate these conversational dynamics is an exciting area of future research for our team.
> > >
> > > We greatly appreciate this interesting and relevant direction for exploration.
> > >
> > > ---
> > > [1] Zhang, X., Xie, H., Li, H., & CS Lui, J. (2020, April). Conversational contextual bandit: Algorithm and application. In Proceedings of the web conference 2020 (pp. 662-672).

---

### Official Review · Reviewer_VTH9 · 2025-06-17

**Clarity:** 1
**Significance:** 2
**Originality:** 3
**Rating:** 3
**Confidence:** 4

**Summary:**

This paper introduces a Thompson Sampling based unified algorithm (SEA) for two distinct LLM alignment scenarios: online interaction (Explore & Exploit setting) and active exploration (Best Arm Identification). The effectiveness of the algorithm is verified across three model scales (1B, 2.8B, 6.9B) and three preference learning algorithms (DPO, IPO, SLiC). The results suggest superior comparative performance both in terms of sample efficiency and win rates w.r.t. reference responses.

**Questions:**

1. Include more validation points with at least one dataset and another model family.
2. Add more technical description for Algorithm 2 (SEA) and please put it in the main paper body.

**Ethical Concerns:**

["NO or VERY MINOR ethics concerns only"]

**Limitations:**

I would suggest an estimate of the energy consumption and the carbon emission for SEA to be included.

**Quality:**

3

**Strengths And Weaknesses:**

Quality: This paper is of good quality as a scientific research paper. The paper poses a scientific question, proposes a solution and then validates its effectiveness. That said, I have some concern about the generalizability of the proposed approach. I think SEA should be validated for another dataset considering this is an empirical study.

Clarity: The paper is very dense with deep mathematical content. I have a few recommendations for the authors to improve the clarity:
(a) Because the central contribution of the paper is an algorithm I would appreciate Algorithm 2 in the main body of the paper in place of many details. (b) Figure 5 is another very important piece of the paper considering it is an empirical study. I think this should be a part of the main body.

Significance: Medium. There are room for improvement. I think the paper should include another family of the model such as LlaMa and Mistral. Also,  it will be good if there is some sense of computational complexity for Algorithm 2. That way the suitability of SEA for the large scale alignment will be established.

Originality: High.

---

> ### Author Rebuttal · Authors · 2025-07-31
>
> Thank you for recognizing the scientific value of our work and its originality, as well as your constructive feedback. We hope to address your concerns below.
>
> ***Q1. Include more validation points with at least one dataset and another model family***
> Thank you for this valuable suggestion. To further validate the generality of SEA, we conducted additional experiments with the following new setting:
> * Base LLM: `meta-llama/Llama-3.2-1B-Instruct`
> * Prompt Dataset: `HuggingFaceH4/ultrafeedback_binarized` (with 60k/2k train/test splits)
> * Reward Oracle: `Skywork-Reward-V2-Llama-3.1-8B`
> * Reward Model Backbone: `Skywork/Skywork-Reward-V2-Llama-3.2-1B`
> * Evaluation temperature: 0.6
>
> This setup uses a different model family (Llama 3) and a more realistic dataset (UltraFeedback). We also leverage a stronger reward oracle (Skywork-V2-8B) and a larger reward model backbone (Skywork-V2-1B).
>
> | step | sea-dpo | online-dpo | sea-dpo-noisy | online-dpo-noisy |
> |------|---------|------------|---------------|------------------|
> |    0 |    0.48 |       0.49 |          0.48 |             0.49 |
> |  100 |    0.59 |       0.51 |          0.54 |             0.50 |
> |  200 |    0.56 |       0.54 |          0.58 |             0.54 |
> |  300 |    0.58 |       0.58 |          0.60 |             0.57 |
> |  400 |    0.60 |       0.59 |          0.62 |             0.58 |
> |  500 |    0.63 |       0.61 |          0.63 |             0.59 |
>
> The table reports win rates against the initial model during training. SEA-DPO consistently learns faster and converges to a higher win rate than Online-DPO, reinforcing the effectiveness and generality of our approach across model families and datasets.
>
> In addition, following reviwer amjx's suggestion, we tested both SEA and online DPO under noisy preference feedback (with 10% chance the preference is flipped). As shown in the table, SEA is robust to feedback noise and shows superior learning efficiency and final performance compared to the baseline.
>
>
> ***Q2. Add more technical description for Algorithm 2, put both Algorithm 2 and Figure 5 in the main body***
>
> Thank you for the helpful suggestion to improve the clarity and accessibility of our method. Due to space constraints in the original submission, we placed Algorithm 2 and Figure 5 in the appendix. In the camera-ready version, we will leverage the additional page allowance to move both into the main body, as recommended.
>
> We will also expand the technical description of Algorithm 2 to clarify its key steps. In particular, the additional computational overhead of SEA compared to vanilla Online DPO arises from two components:
> 1. **Over-sampling candidate responses** for each query to construct duels. This step is embarrassingly parallelizable, and the number of candidates can be tuned to trade off compute cost with labeling/sample efficiency.
> 2. **Inference and training of Epistemic Reward Models**, which quantify uncertainty. These models are lightweight compared to LLM generation, requiring only a single forward pass per sample, and can be further optimized using techniques like LoRA.
>
> Overall, while SEA introduces some additional computation, it remains highly efficient and practical in online learning settings, and we will clarify this in the revised algorithm description.
>
> ***Q3. Include an estimate of the energy consumption and the carbon emission***
>
> Thank you for raising this important point. All experiments were conducted using NVIDIA A100 GPUs. Across all runs (including multiple seeds and ablations), we estimate a total GPU usage of approximately 17,000 GPU hours, which corresponds to an estimated 1,275 kg CO2 emissions, based on the Machine Learning Emissions Calculator [1].
>
> We were mindful of our compute usage and followed standard practices to limit unnecessary runs while ensuring statistical significance. We will include this estimate and the reference to the emissions calculator in the revised version.
>
> ---
> Thank you again for your detailed feedback and suggestions, and we hope our responses and new experiments were able to address any remaining concerns. Please do let us know if you have any further questions as well as what would be expected for score improvement.
>
>
> ---
>
> [1] Lacoste, Alexandre, et al. "Quantifying the carbon emissions of machine learning." arXiv preprint arXiv:1910.09700 (2019).

---

> > ### Comment · Reviewer_VTH9 · 2025-08-05
> > **Thank you**
> >
> > I enjoyed reading your rebuttal. Thanks.
> >
> > Acceptance recommended.

---

> > > ### Author Response · Authors · 2025-08-05
> > > **Thank you again!**
> > >
> > > Thank you for your support!
> > >
> > > We noticed that the review has not yet been updated, and we would be sincerely grateful if you could kindly consider updating it at your convenience to reflect your current assessment.
> > >
> > > Thank you once again!

---

### Official Review · Reviewer_u16V · 2025-07-04

**Clarity:** 4
**Significance:** 2
**Originality:** 2
**Rating:** 4
**Confidence:** 3

**Summary:**

The paper investigates the Online LLM Alignment, which generalises online RLHF and online DPO, by formulating this in a Contextual Duelling Bandit framework.
The authors argue that previous works for LLM Alignment either only use an offline preference dataset or collect online data in an inefficient manner, thus lacking sufficient exploration, which is especially troublesome when the offline dataset is out-of-distribution compared to practice. In section 3, however, the author reviews some works that encourage active exploration through "uncertainty-aware reward models". Hence, in this paper, they propose a Thompson Sampling-based method to directly align the LLM and add exploration for online interactions through posterior sampling.

The key difference here is that they are continuously updating the LLM (and the reward model) throughout the online learning process, to ensure enough data coverage, to address the o.o.d problem, whereas previous methods collect data online based on a fixed policy trained on the offline preference data. On a high level, this approach is quite popular in the Reinforcement Learning field (e.g. DAGGER). I think that Agnihotri, 2024 "Online bandit learning with offline preference data" is a very relevant work to this approach.

The posterior sampling part is approximated using a deep ensemble approach to work with NNs. The authors provide a practical algorithm and demonstrate clearly the effectiveness of their proposal.

**Questions:**

NA

**Ethical Concerns:**

["NO or VERY MINOR ethics concerns only"]

**Final Justification:**

I have read the author's rebuttal and the discussions with other reviewers. No changes required.

**Quality:**

3

**Strengths And Weaknesses:**

The novelty of this paper comes from applying known approaches in the context of LLM online alignment.

---

> ### Author Rebuttal · Authors · 2025-07-31
>
> Thank you for your insightful comments and suggestions to include related work. We will include this relevant paper appropriately in our revision.
>
> We also hope to mention that we have added new experiments with another model family (Llama) and a more realistic dataset (UltraFeedback), and even with _noisy preference feedback_ to address other reviewers' concerns. Please feel free to take a look at the results, which we believe further strengthen the practical applicability of the proposed method.
>
> ---
>
> Thank you again for your insightful feedback, and we hope our responses and new experiments were able to address any remaining concerns. Please do let us know if you have any further questions as well as what would be expected for score improvement.

---

> ### Author Response · Authors · 2025-08-06
> **Thank you!**
>
> Thank you very much for your support!
>
>
>
> Best regards,
>
> The authors

---

### Decision · Program_Chairs · 2025-09-17

**Decision:**

Reject

**Comment:**

This work introduces the notion of dueling bandits for online LLM alignment, building heavily on prior art. While I appreciate the novelty of this application, the authors do not highlight what is specifically missing in prior approaches for this problem class that is closed by the proposed class of techniques. Moreover, there seems to be no new theoretical insights provided by this work. The reviewers have some aesthetic concerns about the breadth of experiments which are inessential relative to this issue at the formulation level.